# Modulation of proteoglycan receptor PTPσ enhances MMP-2 activity to promote recovery from multiple sclerosis

Fucheng Luo[1], Amanda Phuong Tran[2], Li Xin[1], Chandrika Sanapala[2], Bradley T. Lang[3], Jerry Silver[2] & Yan Yang[1,4]

Multiple Sclerosis (MS) is characterized by focal CNS inflammation leading to the death of oligodendrocytes (OLs) with subsequent demyelination, neuronal degeneration, and severe functional deficits. Inhibitory chondroitin sulfate proteoglycans (CSPGs) are increased in the extracellular matrix in the vicinity of MS lesions and are thought to play a critical role in myelin regeneration failure. We here show that CSPGs curtail remyelination through binding with their cognate receptor, protein tyrosine phosphatase σ (PTPσ) on oligodendrocyte progenitor cells (OPCs). We report that inhibition of CSPG/PTPσ signaling by systemically deliverable Intracellular Sigma Peptide (ISP), promotes OPC migration, maturation, remyelination, and functional recovery in animal models of MS. Furthermore, we report a downstream molecular target of PTPσ modulation in OPCs involving upregulation of the protease MMP-2 that allows OPCs to enzymatically digest their way through CSPGs. In total, we demonstrate a critical role of PTPσ/CSPG interactions in OPC remyelination in MS.

---

[1] Department of Neurology, Case Western Reserve University School of Medicine, Cleveland, OH 44106, USA. [2] Department of Neurosciences, Case Western Reserve University School of Medicine, Cleveland, OH 44106, USA. [3] BioEnterprise, 11000 Cedar Avenue, Cleveland, OH 44106, USA. [4] Center for Translational Neurosciences, Case Western Reserve University School of Medicine, Cleveland, OH 44106, USA. These authors contributed equally: Fucheng Luo, Amanda Phuong Tran. These authors jointly supervised this work: Jerry Silver, Yan Yang. Correspondence and requests for materials should be addressed to J.S. (email: jxs10@case.edu) or to Y.Y. (email: yxy33@case.edu)

Multiple sclerosis (MS) is a chronic autoimmune-mediated demyelinating disease characterized by a dramatic loss of clusters of oligodendrocytes (OLs), demyelination, and irreversible neurologic disability[1]. Although remyelination can occur spontaneously, it ultimately fails in regions that develop scar-like, proteoglycan-laden plaques[2]. The underlying mechanisms of failed oligodendrocyte progenitor cell (OPC) differentiation, maturation, and remyelination are still not well understood. Recent studies have identified the regulatory effects of chondroitin sulfate proteoglycans (CSPGs) on OPC maturation and function[3,4].

CSPGs are structural extracellular matrix (ECM) molecules consisting of chains of sulfated glycosaminoglycans (GAGs) attached to a core protein. Upregulation of CSPGs is a hallmark of the scarring process in the CNS and has been well characterized following spinal cord injury[5–8], stroke[9,10], and MS[11–13]. In MS, upregulated CSPGs such as aggrecan and versican have been detected within active demyelinating lesions[13–15]. While permissive laminins promote the spreading, survival, and maturation of OLs[16,17], increased concentrations of CSPGs can outcompete growth-promoting ECM and curtail mouse or human OPC/OL migration, morphological process extension, and maturation[4,18]. CSPG inhibition could be relieved by enzymatic degradation through Chondroitinase ABC to enhance OL maturation in vitro[19]. In vivo, although CSPGs increase temporally in Lysolecithin (LPC)-induced lesions prior to the onset of remyelination[11,20], improved remyelination can occur following CSPG-targeting treatments such as proteoglycan synthesis inhibitors β-d-xyloside[11] or flurosamine[3].

The transmembrane protein tyrosine phosphatase-sigma (PTPσ), and related phosphatase leukocyte common antigen-related (LAR), have been identified as receptors for the inhibitory actions of CSPGs[21,22]. However, the role of PTPσ in OPC/CSPG interactions and MS disease progression is not well understood. A recent study has suggested that ablating or blocking PTPσ may actually exacerbate the course of disease[23] while others have found that knockout of the PTPσ gene stimulates OPCs to increase outgrowth and myelination in vitro despite the presence of aggrecan[4]. Recently, the Silver laboratory has developed a systemically delivered synthetic peptide, Intracellular Sigma Peptide (ISP), that modulates PTPσ and relieves CSPG-mediated inhibition leading to functional recovery following SCI[24–26].

Here, we asked whether ISP treatment would also promote the regeneration of myelin in the setting of demyelinating disease using two different demyelinating mouse models. Indeed, ISP allows OPCs to overcome the inhibitory effects of CSPGs to promote remyelination, as well as robust functional recovery. Further, we demonstrate a mechanism of action underlying CSPG/PTPσ signaling whereby ISP-treated OPCs are stimulated to increase protease activity, especially of MMP-2. We also document that the peptide helps to create a diminished pro-inflammatory environment. In turn, enhanced enzyme production in the context of an altered immune response specifically degrades inhibitory CSPGs that increases OPC migration into and differentiation within demyelinated, CSPG-laden territories. These findings may have strong clinical significance to foster the development of improved CSPG-targeted therapeutic approaches to promote OL regeneration and remyelination within demyelinated lesions in diseases such as MS.

## Results

### Increased CSPGs and PTPσ in demyelinating MS mouse models.
We characterized CSPG expression in demyelinating lesions of MOG$_{35-55}$-induced chronic progressive EAE and LPC-induced acute focal demyelination. Demyelinated EAE and LPC

lesions in the white matter of the spinal cord were visualized with Luxol Fast Blue (LFB) myelin staining (Supplementary Fig. 1). As expected, LFB staining decreased in the lesions of both models. Immunostaining of sections of spinal cord tissue revealed upregulated CSPG expression in demyelinating lesions of EAE- and LPC-afflicted animals compared to vehicle controls (Supplementary Fig. 1). Furthermore, CSPG upregulation progressively increased in the EAE-lesioned spinal cord from 28 to 41 days after immunization (Supplementary Fig. 1A). Tissue sections collected from animals at 7 and 14 days post-LPC injection in the dorsal spinal cord (Supplementary Fig. 1C, D) similarly showed increased production of CSPGs in demyelinating lesions. Increased CSPGs in focally demyelinated areas led us to hypothesize that CSPGs negatively influence OPCs in lesion sites through PTPσ signaling, which may ultimately affect their ability to remyelinate the cord.

CSPGs are known to signal through the receptors PTPσ, LAR[27], and the Nogo receptors 1 and 3[28]. We used a previously published RNA-sequencing transcriptome database of mouse cerebral cortex to search for the gene expressions of PTPRS (PTPσ), PTPRF (LAR), PTPRD (PTPδ), and RTN4R (Nogo receptors) during OPC development[29]. PTPRS gene transcripts (FPKM) were the most abundant type of CSPG receptors in developing OPCs (Supplementary Fig. 2B). Immunostained OPCs/OLs cells derived from wild-type mouse pup brains (postnatal day 1–2) revealed that PTPσ was expressed in the somata and processes of immature Olig2$^+$ and mature CC1$^+$ or myelin basic protein (MBP)$^+$ cells (Supplementary Fig. 2A,C). Western blot analyses also indicated an upregulation of PTPσ in the lesioned spinal cord of EAE- or LPC-induced demyelinating models at day 28 (EAE) and day 7 (LPC) post injections (Supplementary Fig. 2D). In EAE-induced animals, double immunostaining was also performed with antibodies against PTPσ and the OPC marker, Olig2, to reveal increased PTPσ co-labeled with Olig2$^+$ OPCs in demyelinating lesions (Supplementary Fig. 2E). These findings suggest that PTPσ is expressed and upregulated in cells of OL lineages following EAE or LPC-induced disease, and that this receptor presents a tractable target to study the effects of CSPGs and/or receptor manipulations in MS models.

### Modulation of PTPσ allows remyelination and recovery in EAE.
We next tested ISP in an EAE mouse model, which recapitulates chronic progressive demyelination disease processes. Following MOG$_{35-55}$ immunization, animals received intraperitoneal ISP injections (20 μg/mouse, daily) for 41 days at the beginning (EAE ISP Onset) or the peak of sickness (EAE ISP Peak) determined by clinical scoring (Fig. 1a). The control group was injected with 5% DMSO vehicle in parallel. Functional recovery was initially observed in the Onset group after ~10–12 days of ISP administration (i.e., day 23 post immunization). ISP improved clinical scores from 3.5–4 (severe paralysis) to 2–1.5 (limp tail and hind limb weakness). After 20–22 days of ISP treatment (~33 days post immunization), several animals in the Onset group recovered with clinical scores improving to 0.5–1 (limp tail) (Fig. 1b, c). In contrast, control animals remained severely paralyzed with scores remaining around 3.5–4. EAE ISP Peak animals also improved significantly with ISP treatment; however, ISP given at the onset of disease allowed for better recovery (Fig. 1b, c). These improvements were also closely correlated with histological improvements. Lesion sizes were especially reduced in Onset-treated animals as indicated by LFB myelin staining (Fig. 1d, e). Conversely, MBP immunostaining was denser in animals treated with ISP for 41 days compared with control EAE animals (Fig. 1f). Western blotting of MBP protein isoforms also showed restoration of MBP expression in ISP-treated

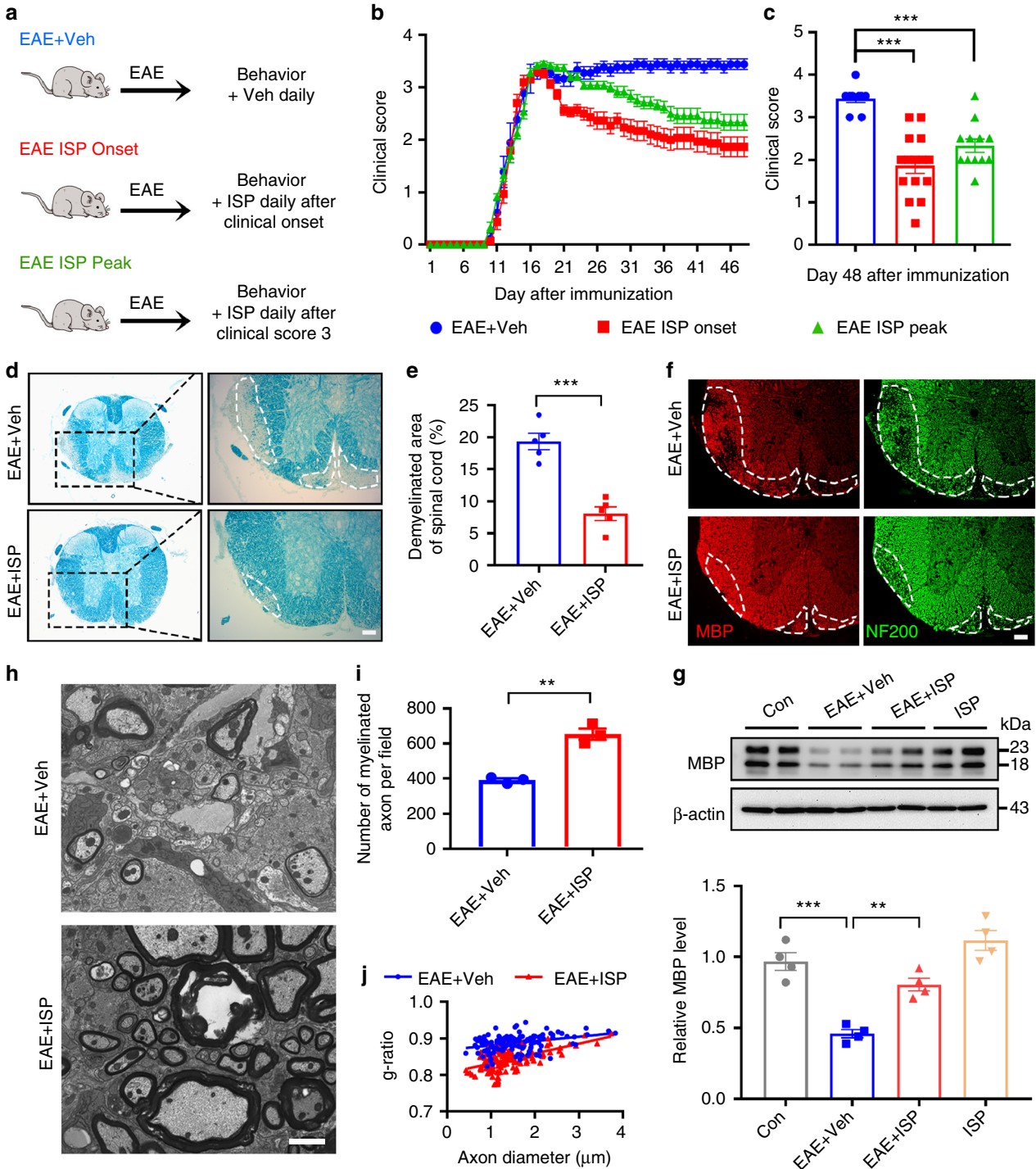

mice (Fig. 1g). Ultrastructural analyses revealed increased myelinated/remyelinated axons in the EAE-lesioned spinal cord in the ISP-treated mice compared to controls (Fig. 1h). Quantitative analysis confirmed an increase of myelinated/remyelinated axons in the ISP-treated group (Fig. 1i) and the G-ratio, which indicates myelination thickness by normalizing the diameter of myelination by axon diameter, was lower in the ISP-treated group compared to the vehicle-treated group (Fig. 1j). Importantly, our results suggest that ISP acted to enhance myelin regeneration rather than prevent demyelination especially since demyelination baselines (LFB) at 18 days following EAE induction were not significantly different between the two groups at this early stage of disease progression (Supplementary Fig. 3E, F).

**ISP leads to less CSPGs and less inflammation in MS lesions**. In addition to observations that CSPGs markedly decreased following ISP treatment of EAE-induced animals (Fig. 2a) we also found altered macrophage dynamics in the same animals. Indeed, macrophages (Iba1) appeared to colocalize with aggrecan (Cat301) especially in the white matter (Fig. 2a), which may be due to activated macrophages depositing or, more likely, phagocytosing aggrecan, which they are not known to produce[30]. We additionally observed decreased Iba1 immunostaining, as well as decreased amounts of aggrecan within Iba1+ macrophages in ISP-treated EAE animals compared to controls (Fig. 2a). We performed further quantification of Iba1 and GFAP at 41 days following EAE induction and saw significant decreases in both

**Fig. 1** ISP promotes functional and histological recovery in EAE mouse model. **a** Diagram of ISP administration to EAE mice at the beginning or the peak of sickness determined by clinical score (Created by F.L. and A.P.T.). **b** Clinical score of disease severity in MOG$_{35-55}$-induced EAE mice treated with ISP or vehicle daily beginning at the onset or the peak of disease. **c** Mean improvement in disease score per animal of EAE cohort in B ($n = 9$ (EAE vehicle group), 15 (EAE ISP onset group) and 12 (EAE ISP peak group), ANOVA F(2,33) = 20.96; Tukey's multiple comparison test, $P_{EAE+Veh\ versus\ EAE\ ISP\ onset} < 0.0001$, $P_{EAE+Veh\ versus\ EAE\ ISP\ peak} = 0.0004$). **d, e** Luxol fast blue (LFB) staining of myelin, demonstrating normal myelin integrity in ISP-treated EAE mice at 48 days post induction, in contrast to the marked loss of myelin present in the spinal cord of vehicle-treated control ($n = 5$ mice/group, $P = 0.0002$, $t = 6.647$, df $= 8$; Two-tailed unpaired Student's $t$ test). Dashed lines demarcate lesion areas. Scale bar $= 100\ \mu m$. **f** Double immunostaining for MBP and neurofilament-200 (NF200) in the thoracic spinal cord of vehicle- and ISP-treated EAE mice at 48 days post induction. Dashed lines demarcate lesion areas. Scale bar $= 100\ \mu m$. **g** Western blot analysis of MBP expression in spinal cord tissue of vehicle or ISP-treated control mice and EAE mice at 48 days post induction. Data are normalized to β-actin protein expression ($n = 4$ mice/group, ANOVA F(3,12) = 26.68; Tukey's multiple comparison test, $P_{con\ versus\ EAE+Veh} = 0.0001$, $P_{EAE+Veh\ versus\ EAE+ISP} = 0.0037$). **h** Electron micrographs from ventral lumbar spinal cords of vehicle and ISP-treated EAE mice 48 days following induction. Scale bar $= 2\ \mu m$. **i** Number of myelinated axons in the spinal cord lesions of vehicle and ISP-treated EAE mice ($n = 3$ mice/group, $P = 0.0014$, $t = 7.815$, df $= 4$; two-tailed unpaired Student's $t$ test). **j** Quantification of the g-ratios (axon diameter/fiber diameter) of myelinated fibers in the ventral lumbar spinal cords of vehicle and ISP-treated EAE mice (EAE + Veh group, g-ratio = 0.8874 ± 0.001901; EAE + ISP group, g-ratio = 0.8423ISP group; $n = 134$ remyelinated axons from three mice/group; $P < 0.0001$, $t = 14.31$, df $= 266$; two-tailed unpaired Student's $t$ test). The data are presented as mean ± s.e.m. *$P < 0.05$, **$P < 0.01$, ***$p < 0.001$

microglia/macrophages and reactive astrocytes, respectively, following ISP treatment (Supplementary Fig. 3A, B). A recent study by Dyck et al.[31] also reported that modulation of LAR family receptor phosphatases with synthetic peptides, including ISP, skewed microglia/macrophages towards an M2 polarization following spinal cord injury. Indeed, immunostaining for markers identifying M1 (iNOS) and M2 (Arginase-1) macrophage polarization revealed supporting evidence that ISP treatment modulates the inflammatory environment in EAE animals (Supplementary Fig. 3C, D). Of note, while microglia/macrophages seem to produce PTPσ after injury, reactive astrocytes do not[25,31].

**ISP enhances myelin repair in LPC-induced demyelination.** We next asked whether ISP modulation of PTPσ-CSPG interactions has similar effects in acute focal demyelination induced by LPC injected into the dorsal column white matter of young adult C57BL6/J mice treated either with ISP (20 μg/day, subcutaneous) or control vehicle starting at 1-day post-LPC injection. After ISP treatment, LPC-induced lesion volumes were significantly reduced at 14 and 21 days post lesion (dpl) compared with vehicle-treated animals as shown by LFB myelin staining (Fig. 3a, b). As indicated in Fig. 3, vehicle-treated animals had an average lesion volume of 1.508 ± 0.069 mm³, 1.035 ± 0.06 mm³, and 0.738 ± 0.027 mm³ after 7, 14, or 21 dpl, respectively. In contrast, ISP-treated animals showed reduced lesion volumes from an average of 1.535 ± 0.058 mm³ at 7 dpl to 0.613 ± 0.043 mm³ at 14 dpl (Fig. 3b). By 21 dpl, we found extensive lesion repair and reduced lesion volumes in ISP-treated mice (1.535 ± 0.058 mm³ to 0.2 ± 0.041 mm³) (Fig. 3b). Immunostaining consistently indicated increased MBP expression in LPC lesions of ISP-treated mice compared with vehicle-treated mice after 14 and 21 dpl (Fig. 3c). Quantitative western blot analysis confirmed increased expression of MBP in LPC-lesioned animals after ISP treatment at 14 dpl (Fig. 3d). Finally, ultrastructural analysis confirmed the number of remyelinated axons in ISP-treated mice at 14 dpl compared to controls (Fig. 3e, f). Consistent with these results, quantitative analyses of the G-ratio between ISP- and vehicle-treated mice revealed increased thickness of myelin sheaths in ISP-treated mice at 14 dpl (Fig. 3g). These experiments showed that ISP treatment accelerated the rate of myelin repair in vivo.

To better visualize the ability of ISP to impact OPCs and subsequent remyelination, we utilized a well-established ex vivo model of myelinating mouse cerebellar slice culture derived from postnatal day 8–10 pups treated with 0.1% LPC for 17–18 h to induce demyelination[32,33]. We found that naive slice cultures developed abundant myelinated axons as shown by MBP and neurofilament (NF200) colocalization (Fig. 4 A, Con). LPC treatment, however, caused profuse demyelination with the production of punctate and disorganized myelin (Fig. 4a, 1 day in vitro (div)). After 8 div, the demyelinated phenotype was still prominent and remyelination was delayed in LPC slices treated with vehicle (Fig. 4a, LPC + Veh, 8 div). In contrast, LPC-demyelinated slices given ISP for 8 div showed increased remyelination (Fig. 4a, LPC + ISP, 8 div) compared to vehicle-treatment. Although increased MBP expression was seen in vehicle-treated slices by 14 div after LPC treatment, the expression of MBP was still disorganized and failed to colocalize well with axons (Fig. 4a, LPC + Veh, 14 div). However, ISP administration for 14 div resulted in abundant remyelinated axons (Fig. 4a, b LPC + ISP, 14 div), which was confirmed with western blot analysis and quantification (Fig. 4c). These slice culture experiments confirm that following LPC treatment, ISP enhances the rate of remyelination perhaps by influencing OPCs directly or possibly microglia derived macrophages as well.

Interestingly, this ISP-enhanced rate of remyelination was correlated with a decrease in aggrecan (Cat301) by 8 div in our cerebellar slice cultures (Supplementary Fig. 4), although aggrecan expression was similar between ISP and vehicle groups at 4 div (Supplementary Fig. 4). By 14 div, we observed simultaneous remyelination through MBP staining and decreased CSPG expression in both groups although ISP-treated slices had enhanced MBP expression and greater CSPG decrease.

To further investigate whether CSPGs are affected by ISP treatment, we double immunostained for MBP and CSPGs in LPC-injected animals. Control LPC-lesioned animals showed upregulated levels of GAG-CSPGs (CS56) and aggrecan (Cat301) that were inversely correlated with decreased expression of MBP at 14 dpl (Fig. 2b). In contrast, ISP-treated animals showed quicker reduction of CSPGs (CS56 and Cat301) and enhanced myelin expression compared to controls (Fig. 2b). It is important to note that while myelin repair does normally occur in LPC-lesioned cerebellar slice cultures, CSPG degradation is much slower than that which occurs after peptide treatment (Supplementary Fig. 4A,B). Thus, we found that ISP treatment not only enhanced the rate of myelin repair, but also was associated with more rapidly decreased CSPG expression over time.

In LPC-injected animals, versican immunostaining was strongest in the penumbra of the lesion where reactive astrocytes (GFAP+) were found (Fig. 2c). This pattern of CSPG deposition confirms recently reported findings by Keough et al.[3] who also reported increased versican secretion by reactive astrocytes. We also examined inflammatory cells and astrocytes in locally demyelinated LPC lesions and altered the timing of ISP treatment to begin our investigation of the mechanism(s) underlying decreased CSPG expression following ISP treatment. Immediately

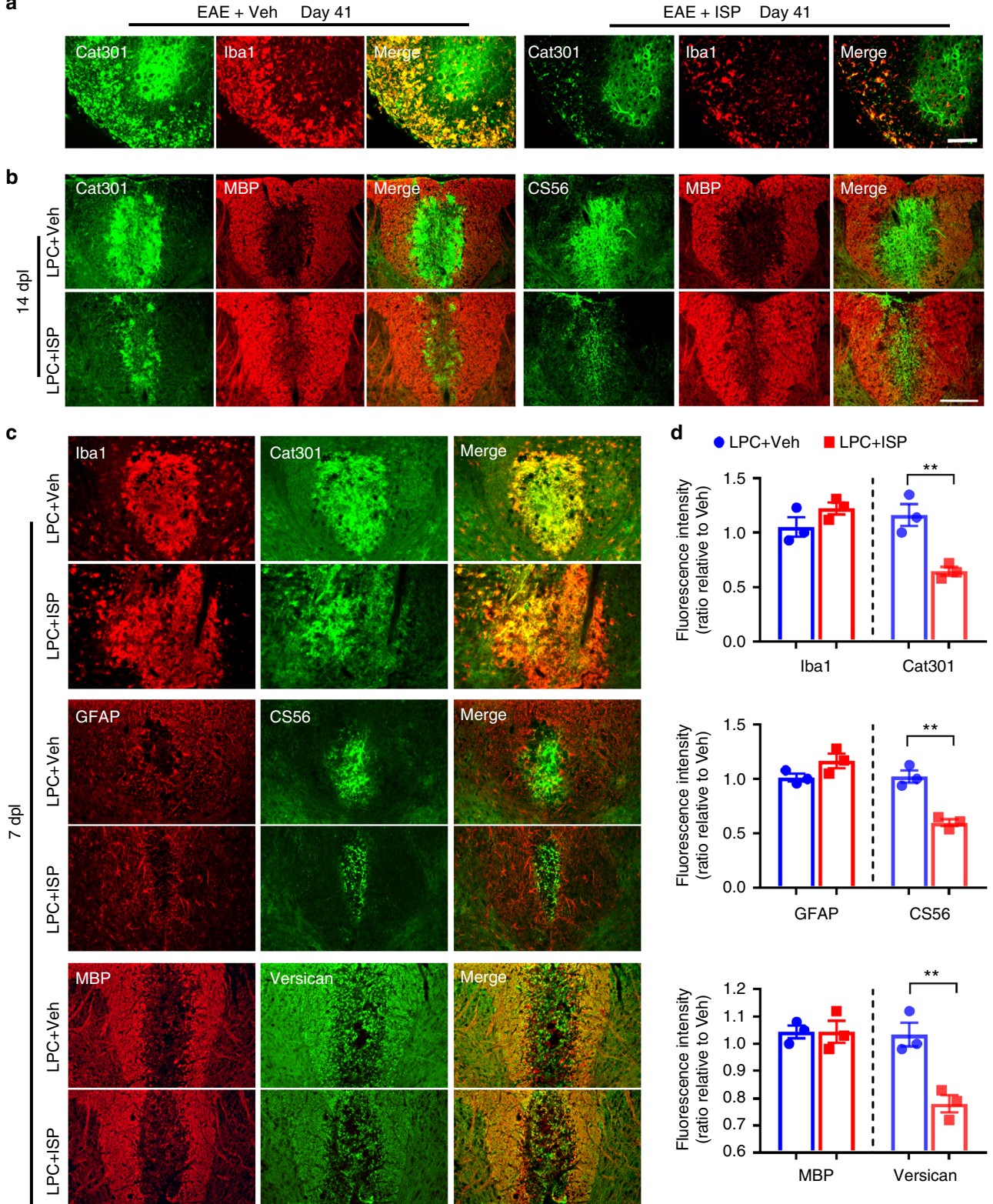

(rather than a 1 day delay) upon LPC injection, mice received ISP (20 µg day/mouse, subcutaneous) for 7 days. Staining of the lesion epicenter revealed no change in the amount of activated microglia (Iba1), reactive astrocytes (GFAP), or MBP myelin protein expression between ISP-treated animals and control groups at this early stage (Fig. 2c, d). This suggests that the extent of LPC-induced injury was initially similar between ISP and control groups. Again, aggrecan staining was colocalized with Iba1⁺

macrophages (Fig. 2c). However, CSPG expression (CS56, Cat301) was significantly reduced after ISP treatment, suggesting that ISP may be involved in the enhanced degradation of CSPGs in demyelinating lesions.

**CSPG loss enhances OPC survival, migration, and maturation.** ISP-induced CSPG disinhibition could result in enhanced

**Fig. 2** ISP decreases chondroitin sulfate proteoglycan (CSPG) load in both EAE and LPC models. **a** Representative immunohistochemistry images of Iba1 and Cat301 (aggrecan-specific antibody) show decreased accumulation of CSPG and microglia/macrophages in the thoracic spinal cord of ISP-treated compared to vehicle-treated EAE mice at 41 days post induction. Scale bar = 100 μm. **b** Representative immunohistochemistry images of MBP, Cat301, and CS56 (glycosaminoglycan specific antibody) show decreased accumulation of CSPG after ISP treatment at 14 dpl in LPC demyelination mice. Scale bar = 100 μm. **c** Representative immunohistochemistry images of Iba1, GFAP, MBP, Cat301, and CS56 show decreased accumulation of CSPG after ISP treatment at 7 dpl in LPC demyelination mice. Scale bar = 100 μm. **d** Relative quantification of immunofluorescence intensity of Iba1, GFAP, MBP, Cat301, and CS56 in the spinal cord of vehicle or ISP-treated LPC mice at 7 dpl ($n = 3$ mice/group, Iba1: $P = 0.1848$, $t = 1.6$, df = 4; Cat301: $P = 0.0092$, $t = 4.719$, df = 4; GFAP: $P = 0.1111$, $t = 2.039$, df = 4; CS56: $P = 0.0028$, $t = 6.55$, df = 4; MBP: $P > 0.9999$, $t = 0$, df = 4; Versican: $P = 0.0095$, $t = 4.669$, df = 4; two-tailed unpaired Student's $t$ test). The data are presented as mean ± s.e.m. **$P < 0.01$

remyelination by regulating OPC proliferation, survival, differentiation, or migration. To distinguish between these possibilities, we began with quantification of proliferating OPCs by immunostaining with Olig2 and Ki67 antibodies in the LPC lesion at 7 dpl. We found that the percentage of Olig2+ OPCs was significantly increased in the lesions of ISP-treated mice compared to vehicle-treated mice (Supplementary Fig. 5A, B), but found no differences in Olig2+/Ki67+ cells between the groups (Supplementary Fig. 5B). To further examine the effects on OPC proliferation by ISP, we peptide-treated cultured OPCs for 2 div on a low concentration of laminin (1 μg/mL) and aggrecan (2 μg/mL) and found no significant differences in proliferation rates between treated and control groups (Supplementary Fig. 5C, D). To investigate whether apoptosis of OPCs was affected by CSPGs, we performed TUNEL staining of OPCs also cultured on low concentrations of laminin and aggrecan and found that ISP treatment decreased the percentage of apoptotic cells (Supplementary Fig. 5E, F). ISP also decreased OPC death when similarly cultured OPCs were challenged with LPC (1 μg/mL, 2 h) (Supplementary Fig. 5E, F).

To examine the effects of ISP treatment on OPC differentiation, we quantified the number of differentiated CC1+ OLs in both EAE animals treated with ISP (41 dpl) and in LPC-injected animals (14 dpl). The percentage of CC1+ OLs was significantly enhanced in lesions of ISP-treated mice compared to controls (Supplementary Fig. 6A–D). ISP-enhanced OPC maturation was also confirmed in vitro with immunopurified OPCs (P1-2 WT mice) cultured on aggrecan and laminin precoated coverslips. Immunostaining of early OPCs (O4+) and mature OLs (MBP+) showed that CSPGs reduced the progressive maturation of OPCs as seen through reduced process lengths of O4- and MBP-expressing cells grown on CSPGs compared to OPCs grown on non-CSPG control substrates (Supplementary Fig. 6E, F). Process outgrowth and maturation of OPCs grown on CSPGs were largely rescued with ISP treatment (quantified by analyzing MBP+ footprints of cells grown on CSPG with or without ISP treatment) (Supplementary Fig. 6E, G). These findings indicate that ISP may be enhancing survival and differentiation, instead of proliferation, of OPCs in demyelinated lesions.

ISP may also be promoting the migration of OPCs into the lesion site where they can survive and subsequently differentiate into their myelinating forms. To explore CSPG/receptor effects on OPC migration, we utilized spinal cord explants derived from P2 WT pups grown on our CSPG gradient spot assay that has been previously used as a potent in vitro model of the inhibitory gradient distribution of CSPGs found in glial scars[25,34]. ISP-treated early (PDGFRα+) and pre-mature (O4+) OPCs derived from the explant were able to cross the CSPG-enriched outer-rim of the gradient spot. In control explants, very few cells were able to migrate across this inhibitory territory (Fig. 5a–c). Thus, in addition to relieving CSPG-related apoptosis and maturational defects, ISP may also be promoting the migration of OPCs into the lesion site where they can survive and subsequently differentiate into their mature myelinating forms. These observations taken together with the reduction of CSPGs in both slice

culture (Supplementary Fig. 4) and in vivo models of MS (Fig. 2) after ISP treatment lead us to hypothesize that targeting PTPσ through ISP induces increased secretion or activation of endogenous proteases.

**ISP enhances protease-dependent enzymatic digestion of CSPGs.** In addition to observations of reduced CSPG expression in ISP-treated ex vivo and in vivo demyelination models, we noticed that ISP-treated PDGFRα+, NG2 + and Olig2 + OPCs left "shadows" of possibly digested GAG-CSPG areas where they infiltrated the aggrecan rim of our spot assays (Fig. 5d, arrows; Supplementary Fig. 7B). The entire outer proteoglycan rim was also reduced in diameter in the presence of ISP (Fig. 5d, compare rim widths). To begin investigating whether protease activity was occurring, we returned to our spot assay to better characterize putative aggrecan degradation. Conditioned media (CM) was collected from immunopurified OPCs treated with vehicle, ISP, or SISP (scrambled ISP) and plated onto freshly made spots. ISP-treated OPC CM significantly reduced CS56 expression compared to vehicle or scrambled peptide controls, as well as a no cell control (Supplementary Fig. 8A–C). Interestingly, the laminin portion of the spot was completely spared as visualized by immunostaining (Supplementary Fig. 8D, E). We also confirmed these results with western blot analyses of OPC CM incubated with aggrecan (20 μg/mL) and laminin (10 μg/mL) collected from OPCs treated with vehicle, ISP, or SISP (Fig. 5e–g).

To independently characterize ISP induction of OPC protease activity, we performed a general enzyme activity assay (EnzChek Kit) based on quenched casein fluorescence and found that ISP treatment of OPCs, indeed, increased protease activity (fluorescence A.U.) compared to vehicle and SISP controls (Fig. 5h, i). Furthermore, ISP increased aggrecan digestion in a dose-dependent manner as visualized through western blot analysis of ISP treatment of OPC CM incubated with aggrecan (Supplementary Fig. 8G, H).

**ISP increases MMP-2 secretion to enhance OPC remyelination.** To begin identifying which critical proteases ISP may be regulating, we incubated vehicle or ISP-treated OPC CM with a protease array blot. We found an increased signal for various enzymes belonging to several classes of proteases (e.g., ADAMTS, Kallikreins, Cathepsins, MMPs) in the ISP-treatment group that are potentially (if produced in sufficient amounts) capable of digesting CSPGs (Supplementary Fig. 7C). Interestingly, three laminin degrading proteases such as Cathepsins L and V and MMP-10 were reduced, suggesting some level of specificity in the regulation of the enzyme cascade that is linked with PTPσ modulation. This result may help explain why we have observed unchanged laminin expression in our ISP-treated in vitro assays (Fig. 5e, g, Supplemantary Fig. 8D, E). To confirm the results from our protease array, we performed western blot analyses of multiple upregulated proteases including MMP-2, 9, and Cathepsin B in ISP or control-treated OPC CM and found that MMP-2 was readily detectable and clearly enhanced after ISP treatment

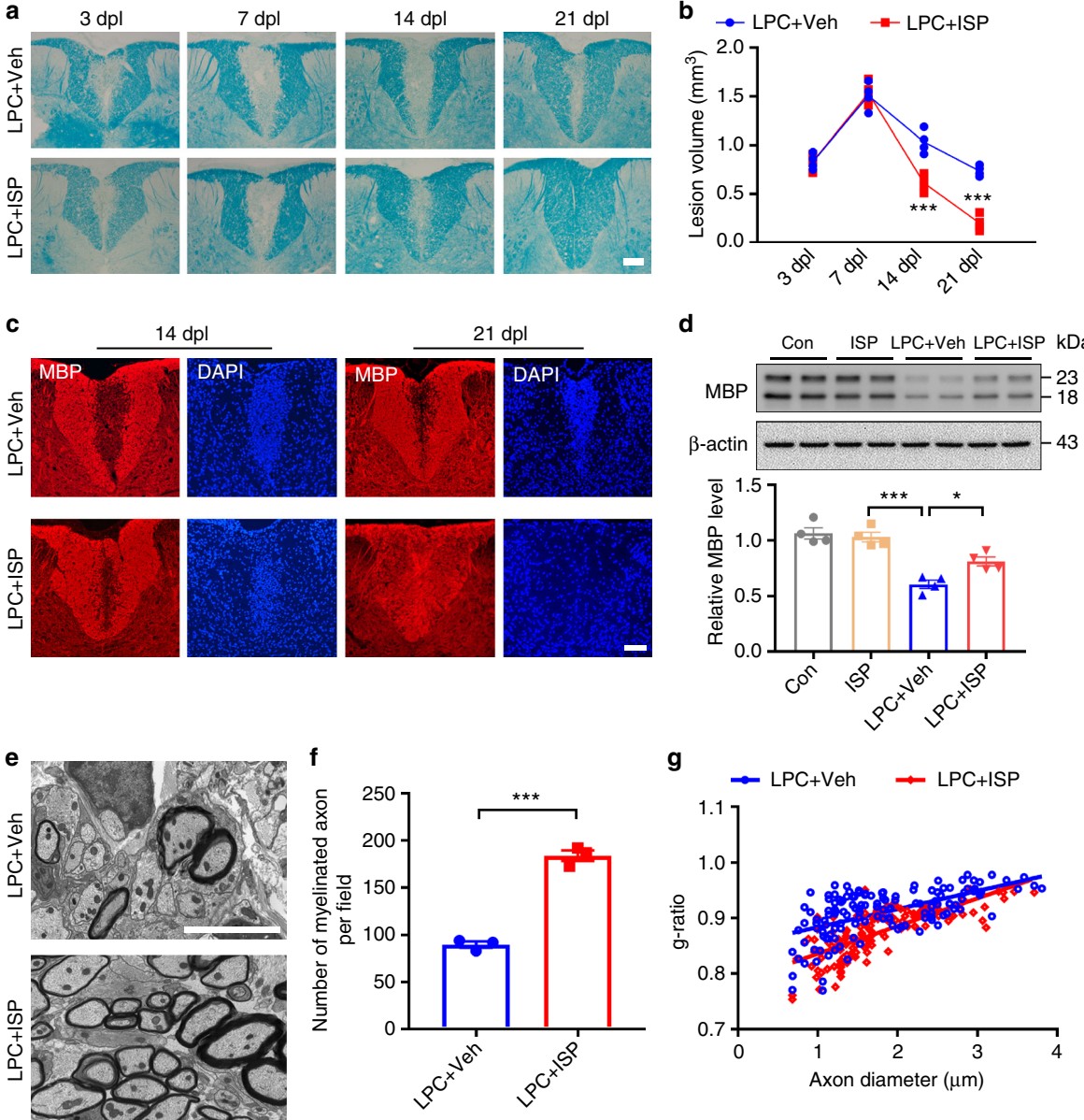

**Fig. 3** ISP promotes remyelination in the spinal cord of lysolecithin (LPC)-demyelinated mice. **a** Representative LFB-stained sections of LPC lesions from the spinal cords of vehicle or ISP-treated mice. Dashed lines demarcate lesion areas. Scale bar = 100 μm. **b** Quantitative analysis of the volume of lesioned spinal cord in vehicle or ISP-treated mice at 3, 7, 14, and 21 dpl ($n = 4$ mice/group, ANOVA F(3,12) = 16.41; Sidak's multiple comparison test, 14 dpl: $P_{LPC +Veh\ versus\ LPC+ISP} = 0.0003$; 21 dpl: $P_{LPC+Veh\ versus\ LPC+ISP} < 0.0001$). **c** Double immunostaining for MBP and DAPI in the spinal cord of vehicle- and ISP-treated mice at 14 and 21 dpl. Dashed lines demarcate lesion areas. Scale bars = 100 μm. **d** Western blot analysis of MBP expression in spinal cord tissue of vehicle or ISP-treated mice at 14 dpl. Data are normalized to β-actin protein expression ($n = 4$ mice/group, ANOVA F(3,12) = 24.21; Tukey's multiple comparison test, $P_{con\ versus\ LPC+Veh} < 0.0001$, $P_{LPC+Veh\ versus\ LPC+ISP} = 0.0276$). **e** Representative electron microscopy images of LPC lesions from the spinal cord of vehicle or ISP-treated mice at 14 dpl. Scale bar = 5 μm. **f** The number of myelinated axons in LPC-induced lesions from vehicle or ISP-treated mice at 14 dpl ($n = 3$ mice/group, $P = 0.0001$, $t = 14.26$, $df = 4$; two-tailed unpaired Student's $t$ test). **g** The myelin g-ratio in the LPC lesions of vehicle or ISP-treated mice at 14 dpl (LPC + veh group, g-ratio = 0.9103 ± 0.003583; LPC + ISP group, g-ratio = 0.8741 ± 74103599; $n = 139$ remyelinated axons from 3 mice/group; $P < 0.0001$, $t = 7.142$, $df = 276$; two-tailed unpaired Student's $t$ test). The data are presented as mean ± s.e.m. *$P < 0.05$, **$P < 0.01$, ***$p < 0.001$

(Fig. 6c, d). Staining of cultured OPCs showed that MMP-2 is expressed in O4$^+$ OPCs within their processes and appears to intensify after peptide treatment (Fig. 6h).

To confirm that OPC CM-derived MMP-2 activity is enhanced by ISP treatment, we performed gelatin zymography and found that MMP-2 gelatin-degrading activity was significantly increased upon ISP treatment over controls (Fig. 6a, b). MMP-9 activity was barely visible by gelatin zymography (Fig. 6a). We also blotted for MMP-10, a protease that degrades fibronectin, laminin, and

elastin, and found that it is secreted in far lower amounts than MMP-2 (Fig. 6c). Vehicle, ISP, and SISP-treated OPC cultures seem to secrete MMP-10 in equally low amounts (Fig. 6c, e), suggesting that enhanced MMP-2 secretion by ISP may be specific to PTPσ modulation. In addition to ISP-treated OPC CM, OPC lysates were also analyzed by western blot and showed a decrease in MMP-2 expression normalized over GAPDH loading control suggesting that MMP-2 secretion may be enhanced by ISP (Fig. 6f, g). To test this, we incubated aggrecan with OPC CM

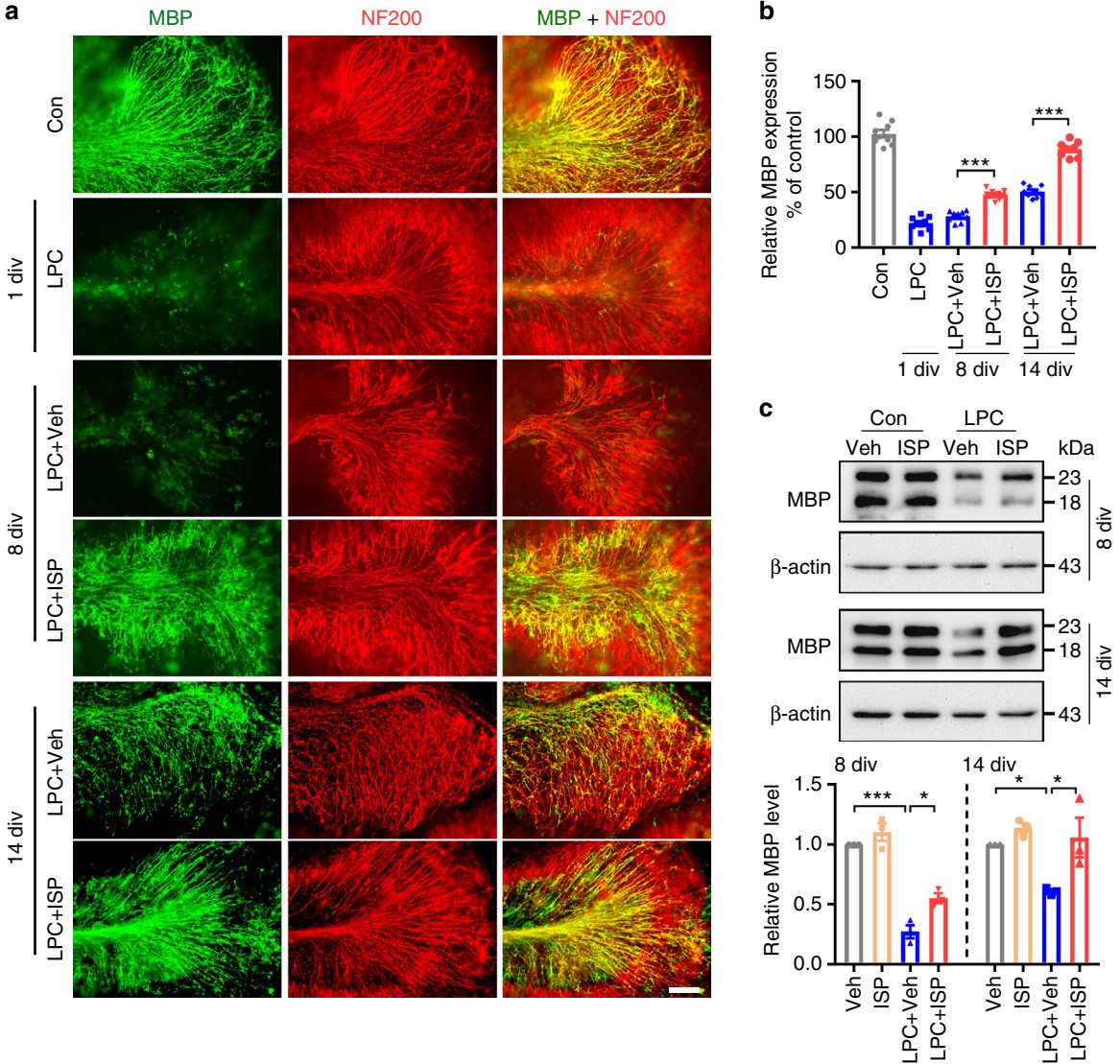

**Fig. 4** ISP accelerates remyelination in LPC-treated organotypic cerebellar cultures. **a** Representative immunohistochemistry images of MBP and neurofilament-200 (NF200) show normal myelination in naive (Con) sections, LPC-induced demyelination at 1 day in vitro (div), and increased remyelination after ISP treatment in LPC-demyelinated cerebellar slices at 8 div and 14 div. Scale bar = 100 μm. **b** Bar graph illustrates relative MBP immunoreactivity (i.e., colocalization of MBP and NF200) in cerebellar slices compared to no treatment (100% as control) (n = 9 slices from 3 independent replicates per group, ANOVA F(5,48) = 230.4, Tukey's multiple comparison test, 8 dpl: $P_{LPC+Veh \text{ versus } LPC+ISP}$ < 0.0001, 14 dpl: $P_{LPC+Veh \text{ versus } LPC+ISP}$ < 0.0001). **c** Western blot analysis of MBP expression in vehicle or ISP-treated cerebellar slices at 8 div and 14 div. Data are normalized to β-actin protein expression (n = 3 independent replicates per group. 8 div: ANOVA F(3,8) = 58.89, Tukey's multiple comparison test, $P_{Veh \text{ versus } LPC+Veh}$ < 0.0001, $P_{LPC+Veh \text{ versus } LPC+ISP}$ = 0.0187; 14 div: ANOVA F(3,8) = 6.281, Tukey's multiple comparison test, $P_{Veh \text{ versus } LPC+Veh}$ < 0.048, $P_{LPC+Veh \text{ versus } LPC+ISP}$ = 0.025.) The data are presented as mean ± s.e.m. *P < 0.05, **P < 0.01, ***p < 0.001

treated with an exocytosis inhibitor, Exo1 (10 μg/mL), in conjunction with ISP. At sufficient concentrations, Exo1 has been observed to reversibly inhibit exocytosis through its inhibition of the Arf GTPase[35]. We found that Exo1 partially rescued aggrecan GAG digestion (Fig. 6i, j). We also performed the same experiment with a broad MMP inhibitor, GM6001 (25 μM), and a specific MMP-2 inhibitor (OA-Hy, Calbiochem, 100 nM) with ISP and found that GAG digestion was partially rescued in both cases indicating that ISP-induced CSPG degradation may very well be perpetrated by the metalloprotease family and MMP-2 predominantly (Fig. 6I, J). GM6001 and an MMP-2 inhibitor additionally rescued CS56 spot degradation (Supplementary Fig. 9A–D).

We returned to the spot assay to test whether MMP inhibition decreases OPC migration across the CSPG rim. Treatment of

OPCs with GM6001 (25 μM) and the specific MMP-2 inhibitor (OA-Hy, 100 nM) effectively halted OPC entry into the CSPG-rich area even in the presence of ISP (Fig. 7a, b). This suggests that ISP-induction of enhanced OPC migration may be dependent on MMPs. Finally, to test the functional necessity of MMPs on remyelination, we treated LPC-demyelinated cerebellar slices with GM6001 or the specific MMP-2 inhibitor in conjunction with ISP. Colocalization of MBP and neurofilament [+] axons was, indeed, decreased with GM6001 and the MMP-2 inhibitor treatment despite the presence of ISP (Fig. 7c, d). Furthermore, 2 div inhibition of MMP-2 in LPC-challenged (1 μg/mL, 2 h) OPCs cultured on low concentrations of aggrecan and laminin ablated the survival-promoting effects of ISP as assessed through TUNEL staining (Supplementary Fig. 9E). The MMP-2 inhibitor, however, did not seem to increase apoptosis

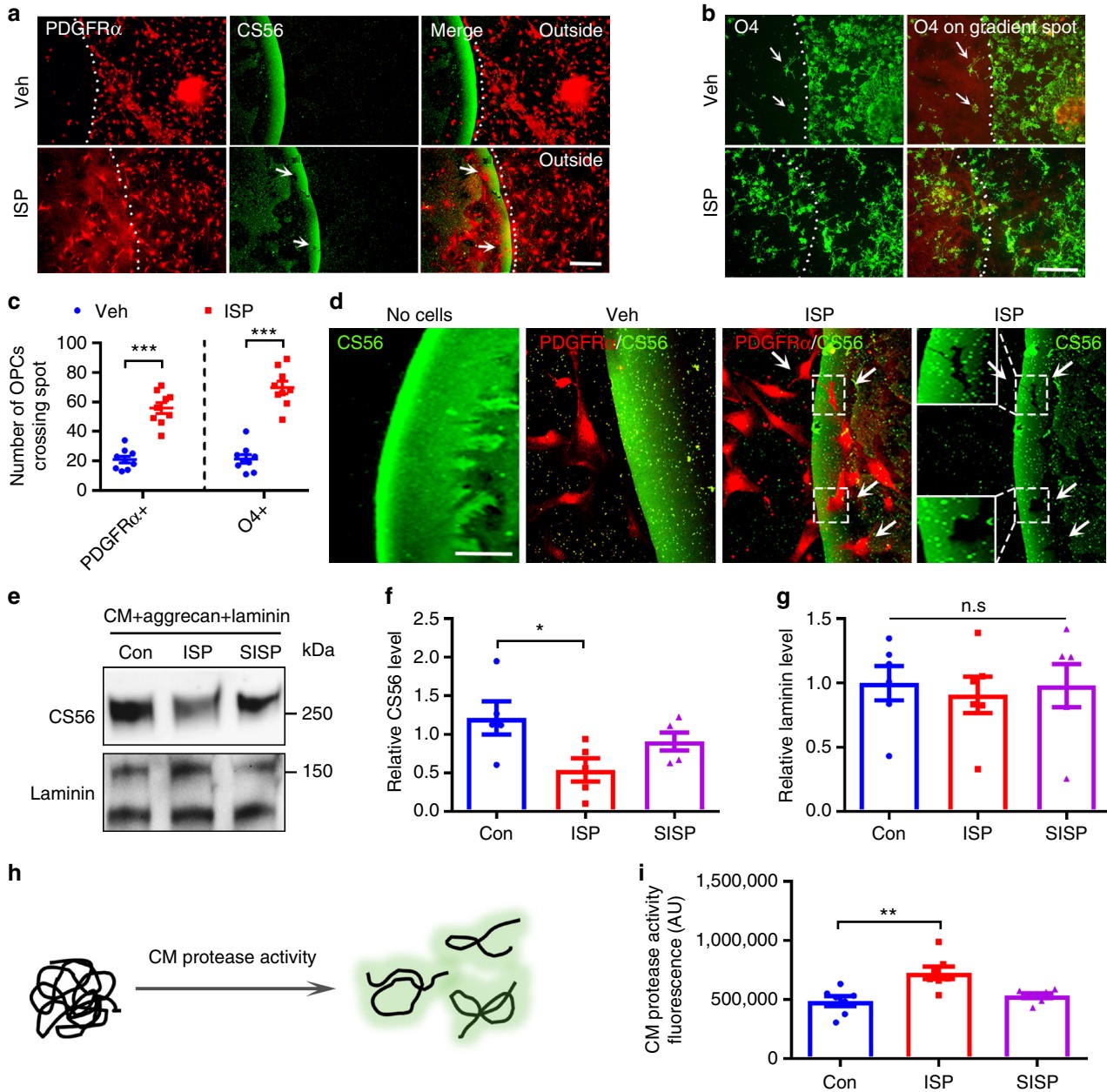

**Fig. 5** ISP increases CSPG-degrading protease activity. **a**, **b** CSPG gradient crossing assay shows that ISP treatment promotes the crossing of PDGFRα+ or O4+ OPCs through the gradient of CSPG. Scale bar = 100 μm. **c** Quantification of immunostaining for the amount of PDGFRα+ or O4+ OPCs crossing the CSPG barrier after vehicle or ISP treatment (n = 9 spots from 3 independent replicates. PDGFRα: $P < 0.0001$, $t = 7.99$, $df = 16$; O4: $P < 0.0001$, $t = 9.419$, $df = 16$, two-tailed unpaired Student's $t$ test). **d** Representative immunostained images of CS56 and PDGFRα+ OPCs on CSPG barrier depicting CSPG degradation after ISP treatment as they cross the barrier to leave CS56 "shadows" (inset and arrows). Scale bar = 50 μm. **e** To investigate protease activity, OPC conditioned media (CM) was treated with vehicle control or 2.5 μM ISP or scrambled ISP (SISP) and incubated with aggrecan (20 μg/mL) or laminin (10 μg/mL), then analyzed through western blots. **f** Quantification of glycosaminoglycan moiety through CS56 immunoblotting reveals significant CS56 degradation following ISP treatment (One-Way ANOVA, Dunnett's posthoc test, $P = 0.0432$, $F(2,12) = 4.131$, $N = 5$ western blots). **g** Quantification of laminin immuoblotting shows no significant changes (One-Way ANOVA, Tukey's posthoc test, $P = 0.9024$, $F(2,15) = 0.1034$), $N = 6$ western blots). **h** Quenched casein in EnzChek protease activity assay fluoresces once it becomes cleaved. **i** Quantification of EnzChek protease activity assay reveals significant protease activity in OPC CM treated with 2.5 μM ISP over control(One-Way ANOVA Dunnett's posthoc test, $P = 0.0015$, $F(2,18) = 9.534$, $N = 26$ from 7 replicates). The data are presented as mean ± s.e.m. *$P < 0.05$, **$P < 0.01$, ***$p < 0.001$. n.s. not significant

compared to vehicle control on aggrecan/laminin alone (Supplementary Fig. 9E). Specific Inhibition of MMP-2 additionally negated gains in MBP footprints of mature OLs grown in the presence of ISP on low concentrations of aggrecan/laminin (Supplementary Fig. 9F, G).

To further elucidate the necessity of MMP-2 activity following ISP treatment in enhancing remyelination, we utilized a lentiviral particle-delivered shRNA construct. We first validated this shRNA approach using lentiviral delivery as well as western analysis to knockdown MMP-2 in OPC cultures infected for 48 h (Supplementary Fig. 10A). shRNA knockdown of MMP-2 was able to reduce the area of extended MBP+ processes of OLs cultured on aggrecan in vitro (Supplementary Fig. 10B, C) and the ability of OPCs to migrate past a high aggrecan barrier

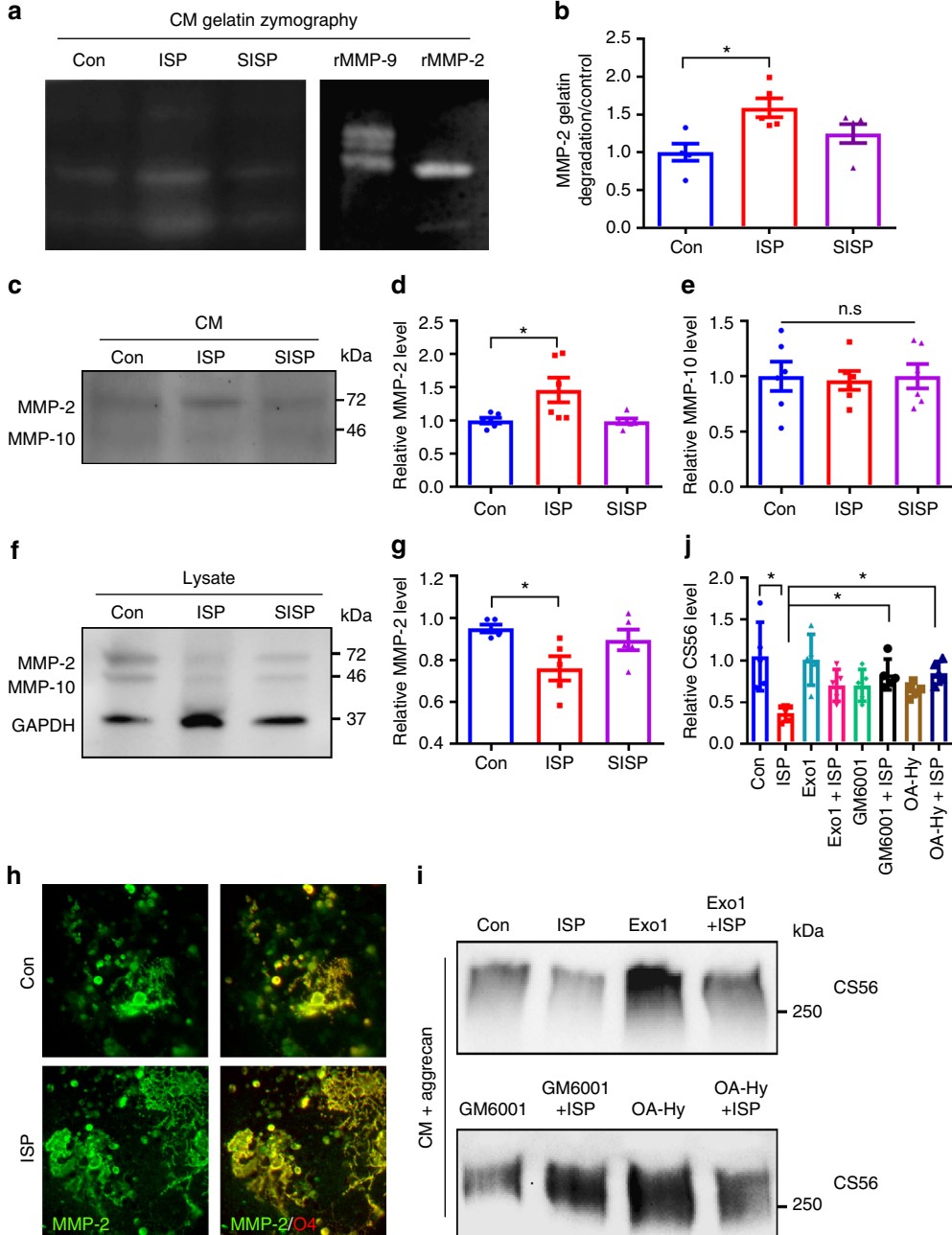

**Fig. 6** ISP increases MMP-2 secretion and activity. To further characterize protease activity, **a**, **b** cultured OPCs were treated with vehicle control or 2.5 µM ISP or SISP, concentrated, then loaded onto gelatin SDS/PAGE gels for zymography analysis. 25 ng of recombinant MMP-9 or MMP-2 served as positive controls. Quantification of active MMP-2 lanes of gelatin zymography reveals significant MMP-2 activity following ISP treatment over control (One-Way ANOVA, Dunnett's posthoc test, $P = 0.0161$, $F(2,12) = 5.937$, $N = 5$ zymograms). **c** Enhanced MMP-2 expression following ISP treatment was confirmed in western blots of concentrated OPC conditioned media (CM). **d** Quantification of MMP-2 immunoblotting was significantly enhanced following ISP treatment over control (One-Way ANOVA, Dunnett's posthoc test, $P = 0.0150$, $F(2,15) = 5.63$, $N = 6$ western blots). **e** MMP-10 immunoblotting was not significant among treatments (One-Way ANOVA, Tukey's posthoc test, $P = 0.9619$, $F(2,15) = 0.03899$, $N = 6$ western blots). **f**, **g** Western blot analysis showed a decrease in MMP-2 expression normalized over GAPDH loading control in OPC lysates after ISP treatment. **h** Immunostaining of O4[+] (red) OPCs shows MMP-2 concentrated in OPC soma and processes. Scale bar = 100 µm. **i**, **j** To explore whether ISP induces secretion of proteases to degrade CSPGs, cultured OPCs were treated with the following drugs with or without 2.5 µM ISP: exocytosis inhibitor Exo1 (10 µg/mL), general metalloprotease inhibitor GM6001 (25 µM), or specific MMP-2 inhibitor (OA-Hy, Calbiochem, 100 nM). Collected CM was incubated with aggrecan (20 µg/mL) and immunoblotted with CS56. Quantification of CS56 reveals significant ISP-induced degradation of CSPGs over control, Exo1 + ISP, GM6001 + ISP, and OA-Hy + ISP (One-Way ANOVA, Tukey's posthoc test, $P = 0.0010$, $F(7,32) = 4.749$, $N = 5$ western blots). The data are presented as mean ± s.e.m. *$P < 0.05$. n.s. not significant

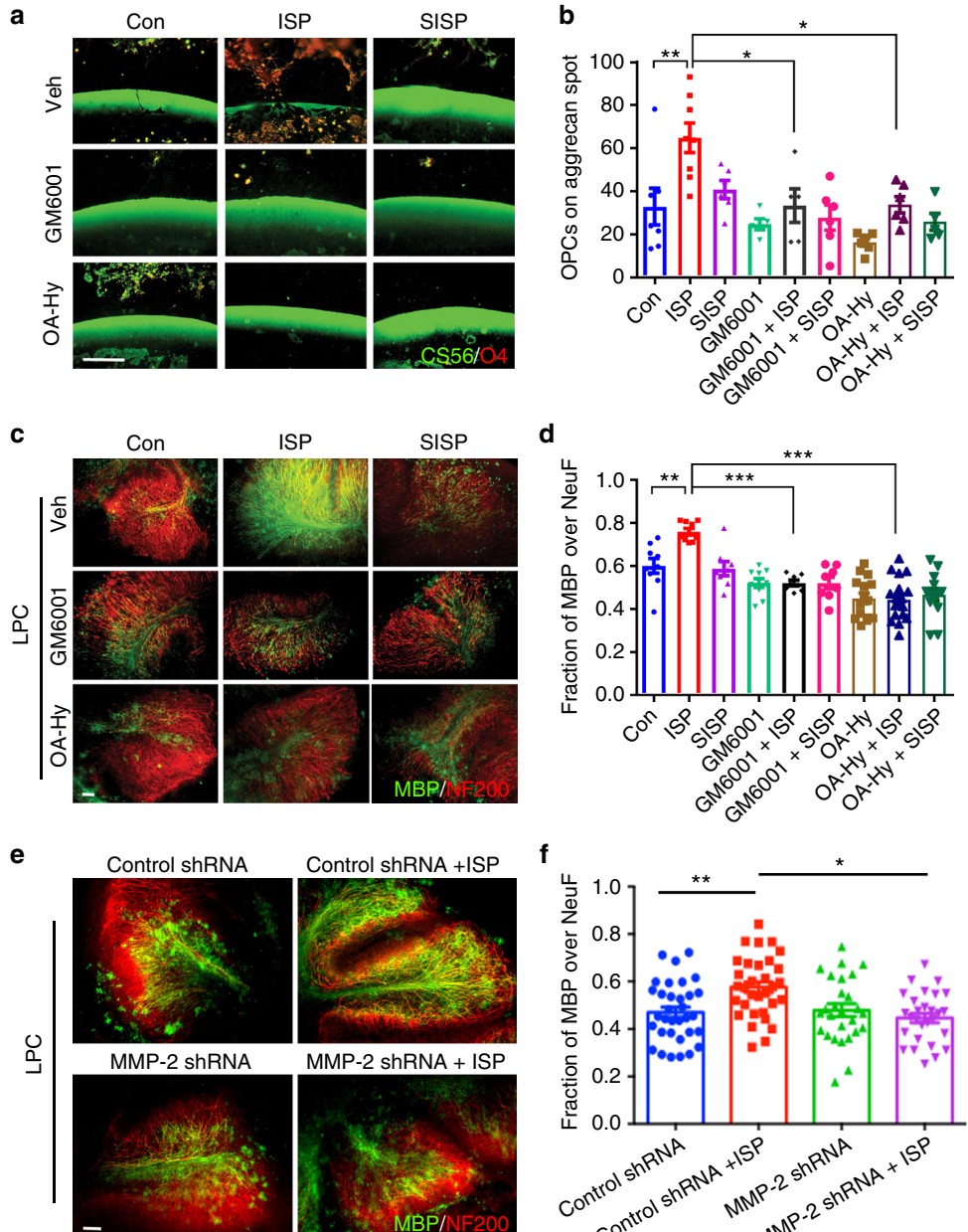

**Fig. 7** ISP-induced MMP-2 activity increases OPC migration and remyelination through CSPG degradation. **a** To ask whether ISP-induced protease activity is involved in OPC migration and remyelination, cultured OPCs were plated onto coverslips with CSPG spot gradients and treated with vehicle control, 2.5 μM ISP or SISP, 25 μM GM6001±ISP or SISP, or 100 nM MMP-2 inhibitor (OA-Hy)±ISP or SISP. Scare bar = 100 μm. **b** The amount of O4+ OPCs crossing the CSPG barrier was counted and quantified. ISP treatment (N = 37 spots, 5 replicates) significantly induced greater O4+ OPC migration past the CSPG barrier compared to control (One-Way ANOVA, Tukey's posthoc test, P = 0.0002, F(11,48) = 5.013), GM6001 + ISP (N = 20 spots), or OA-Hy + ISP (N = 20 spots) treatments. N(Spots) = 31 Control, 26 SISP, 18 GM6001, 20 GM6001 + SISP, 18 OA-Hy, 19 OA-Hy + SISP). **c** To test whether remyelination was affected by protease inhibition, P7-9 cerebellar slices were all treated with LPC for 18 h, then treated with control vehicle, 2.5 μM ISP or SISP, 25 μM GM6001± ISP, or 100 nM OA-Hy±ISP for 9 days before staining for neurofilament (NF200) or MBP. Scale bar = 100 μm. **d** Remyelination was quantified through MBP and neurofilament colocalization. ISP treatment (N = 35 images from 4 replicates with up to 13 sections total) significantly increased MBP-neurofilament colocalization over control (One-Way ANOVA, Tukey's posthoc test, P = 0.0001, F (8, 88) = 13.61), GM6001 + ISP (N = 28), and OA-Hy + ISP (N = 23) groups. N(images) = 17 Control, 22 SISP, 48 GM6001, 20 GM6001 + SISP, 22 OA-Hy, 22 OA-Hy + SISP. **e**, **f** Cerebellar slice cultures were treated with lentiviral constructs for 48 h before LPC treatment. Vehicle or ISP (2.5 μM) treatment followed for 6 days in vitro. Scale bar = 100 μm. MBP immunofluorescence (green) was then quantified with NF200 (red) (One-Way ANOVA, Tukey's posthoc test. P = 0.0005. F(3, 116) = 6.395. N = around 30 images from 10 slices). The data are presented as mean ± s.e.m. *P < 0.05, **P < 0.01, ***p < 0.001

(Supplementary Fig. 10D, E) despite ISP treatment. As expected, shRNA knockdown of MMP-2 also attenuated ISP-induced remyelination in cerebellar slice cultures (Fig. 7e, f).

We next characterized MMP-2 expression in vivo using immunohistochemistry and found that while the dorsal column of the naive spinal cord expressed a baseline of some MMP-2 protein, LPC injection into the same area somewhat increased MMP-2 expression at 7 dpl (Fig. 8a, b). However, ISP treatment markedly enhanced MMP-2 expression in the LPC-injected site, which was also confirmed with western blot analysis of the

affected spinal cord areas (Fig. 8c). Immunostaining of ISP-treated LPC-demyelinated cords showed MMP-2 colocalizing with Olig2-identified OPCs (Fig. 8d), but also with Iba1-labeled immune cells (Fig. 8e). To explore further the necessity of MMP-2 activity in ISP-enhanced remyelination, we returned to the LPC model and analyzed lesion volume following myelin staining at 18 dpl (Fig. 8f, g) with an MMP-2 inhibitor (OA-Hy) or shRNA construct targeting MMP-2 delivered with lentiviral particles. Interestingly, pharmacological inhibition of MMP-2 increased the lesion volume over vehicle control suggesting that baseline levels of MMP-2 may be facilitating slow remyelination in this model. The addition of MMP-2 pharmacological inhibition or MMP-2 shRNA-mediated knockdown attenuated the enhanced remyelinating effects of ISP which correlated with a concomitant increase in CS56 immunoreactivity (Supplementary Fig. 11A, D), decreased accumulation of $Olig2^+$ OPCs (Supplementary Fig. 11B, E), and reduced expression of MBP protein in the lesion epicenter (Supplementary Fig. 11C, F). Together, these results indicate the importance of PTPσ-regulated MMP-2 secretion by OPCs, but also possibly microglia, to assist them in their migration and ability to remyelinate despite high CSPG deposition after focal demyelinating injury.

## Discussion

We have elucidated a critical role for the CSPG receptor PTPσ following its modulation in the restoration of OPC homeostasis in a variety of MS models where proteoglycan deposition during lesion associated scar formation potently inhibits OPC migration, differentiation, and remyelination. Here we report that targeting PTPσ with a systemically delivered peptide enhances the rate of myelin repair in LPC-induced lesions and stimulates robust myelin regeneration and functional recovery following chronic demyelinating EAE. We also present a finding linking PTPσ modulation with altered immune polarization and enhanced protease activity. This underscores the important role that CSPGs play following CNS demyelinating diseases and identifies a strategy that can target MS lesions broadly throughout the neuraxis to relieve CSPG-mediated inhibition.

The downstream mechanisms following peptide modulation of the receptor PTPσ that allow cells to overcome CSPG inhibition are largely unknown. Do treated cells simply fail to recognize the inhibitory chondroitin GAG chains or do they selectively modify the substrate to remove only inhibitory components of the ECM, sparing those that promote their growth? Protease activity is heavily regulated at several levels including transcription, translation, secretion, localization, activation, and inhibition to prevent unfettered, potentially damaging activity[36]. Left uncontrolled, proteases are able to degrade a broad range of proteins with potentially disastrous outcomes as seen in "protease storms" following a variety of CNS injuries. In the relapsing phase of MS, nonspecific protease upregulation associated with rampant inflammation and myelin degeneration have been well characterized[37]. However, the beneficial effects of finely regulated protease secretion following injury to promote tissue repair are becoming more appreciated. Here, we present a finding linking PTPσ modulation with enhanced MMP-2 protease activity by OPCs, which aids in their digestion through CSPG-laden demyelinated plaque that envelops the MS-like lesion. Through in vitro, ex vivo and in vivo assays, we have identified the necessity of MMP-2 activity through PTPσ modulation not only for OPC migration but also for improved OPC survival, maturation, and remyelination. MMP-2 upregulation has been previously identified to allow stem cells to invade CSPG-containing regions of the glial scar[38] and improve remyelination of peripheral axons by Schwann cells in culture[39].

Interestingly, previous work on damaged peripheral nerves, where regeneration occurs spontaneously has also noted the fine-tuned regulation of neuronal or Schwaan cell MMP-2-induced degradation of CSPGs but with laminin sparing[40]. We have also observed laminin sparing and concomitant CSPG degradation with ISP treatment, which may be one mechanism by which OPCs are able to infiltrate into and survive otherwise CSPG-dense demyelinated lesions. This highlights the precise regulation of proteases by OPCs, and possibly immune cells, through PTPσ to promote controlled CSPG degradation. Whether MMP-2 plays other roles in OPC homeostasis independent of CSPG disinhibition will need to be further explored.

Importantly, while we were able to detect MMP-2 in our various assays, we cannot exclude the possibility that other proteases may also be activated either in parallel or along the same MMP-2 signaling axis. Other proteases that could be involved might include a variety of other MMPs such as MMP-12, which has been shown to regulate OPC maturation and morphological differentiation[41]. Although found at very low concentrations in our assays, MMP-9, which has been shown to facilitate remyelination[42], may be upregulated as OPCs mature. Additionally, aggrecan-degrading ADAMTS4 has recently been found to be restricted to mature OLs and implicated in efficient myelination[43]. In fact, knockout of ADAMTS4 seems to impair myelination enough to impair motor function in mice. We have recently shown that proteoglycan degrading Cathepsin B is upregulated in sensory and serotonergic neurons by ISP treatment, which markedly improves functional recovery following contusive spinal cord injury. This suggests that protease regulation may be a fundamental downstream target of the PTPσ receptor[44].

Recently, Dyck et al.[31] found that daily treatment of spinal cord injured rats with our LAR family receptor blocking peptides dampened the pro-inflammatory environment induced by traumatic injury. Indeed, peptide-treated macrophages from injured cords were found to express more reparative phenotype M2 markers and anti-inflammatory factors such as IL-10. Work from the Bradbury group involving large-scale CSPG-degrading Chondroitinase ABC therapies following spinal cord injury also skewed macrophages toward the M2 spectrum[45] again with enhanced IL-10 secretion[46]. Our study begins to confirm this observation in a peptide-treated EAE model of MS where we observed decreases in CSPG load, as well as the destructive M1 macrophage phenotype marker iNOS and increases in M2 associated Arginase-1.

What might be the further effect of modulating the inflammatory environment by ISP on the behavior of microglia/macrophages or the evolution of CSPGs in the MS lesion? Why do we see decreased numbers of microglia/macrophages and/or less aggrecan within macrophages over time after peptide treatment? It has been shown that incubation of microglia/macrophages with CSPG-secreting reactive astrocytes[47] or CSPGs in solution or bound to a substrate[31,48] curtails the rate at which they phagocytose debris and degrade extracellular proteins, such as β-amyloid, in vitro. Our study suggests, in conjunction with Dyck et al.[31], that ISP may be markedly enhancing microglial phagocytic/digestive capacity. By modulating the cognate receptor of CSPGs, OPCs, and microglia may be working together to more rapidly clear inhibitory CSPGs and other cellular remnants. Thus, in altering the pro-inflammatory environment toward an M2 state[49] and inducing focal protease activity by selective cells, ISP may provide additional CSPG disinhibition, culminating in myelin regeneration. Whether modulation of PTPσ by peptide treatment additionally enhances phagocytic or protease activity in $Iba1^+$ cells in the setting of MS to further aid in CSPG clearance, as well as cellular debris removal in general, will require further investigation.

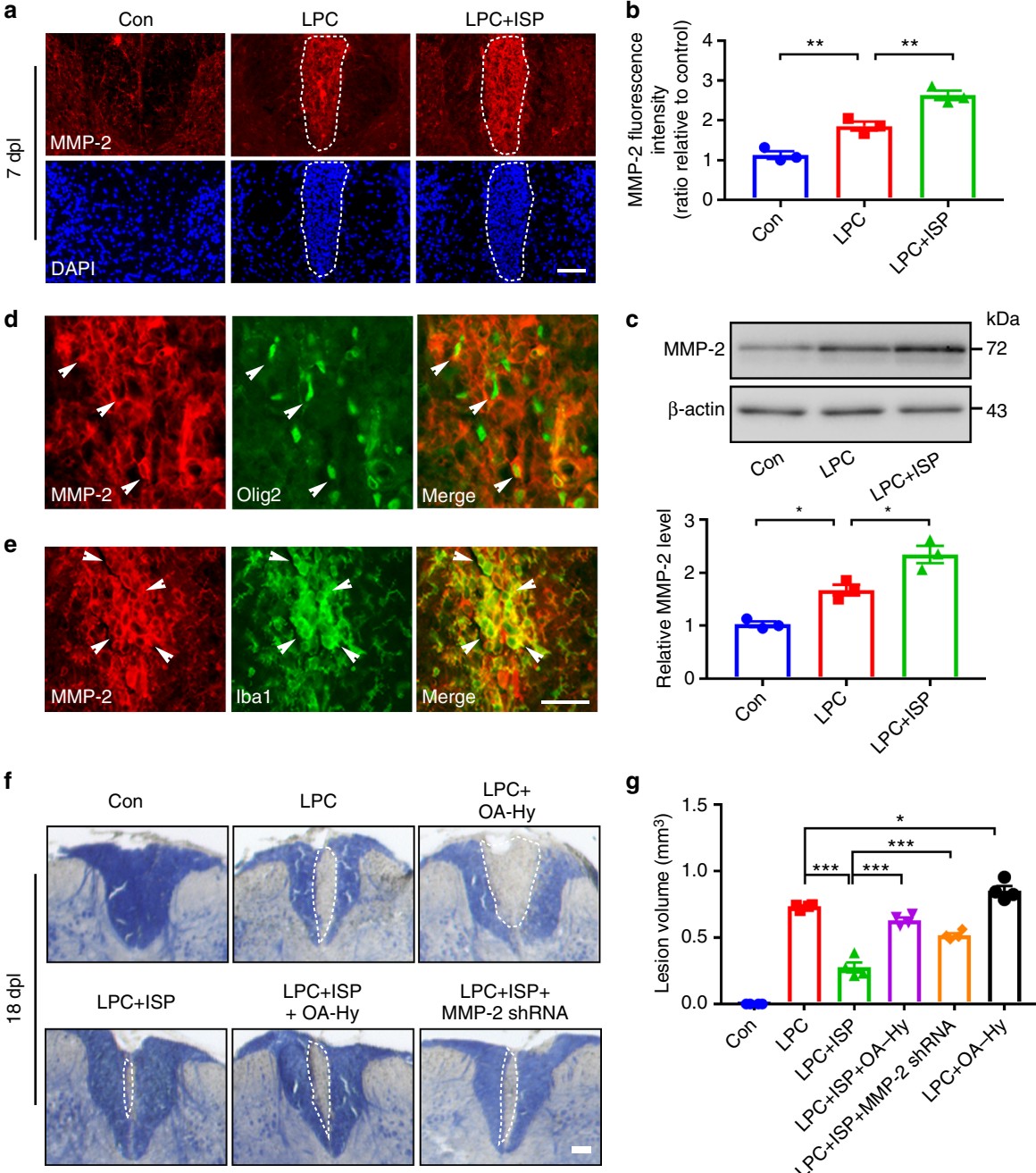

**Fig. 8** ISP promotes myelin repair through increasing MMP-2 expression in LPC-induced demyelination model. **a** Representative images from immunohistochemistry of MMP-2 and DAPI show increased levels of MMP-2 in the spinal cord of ISP-treated mice at 7 days post-LPC injection (dpl) compared to naive or LPC vehicle control cords. Dashed lines demarcate lesion areas. Scare bar = 100 μm. **b** Relative quantification of immunofluorescence intensity of MMP-2 in the spinal cord of ISP-treated mice at 7 dpl ($n = 3$ mice/group, ANOVA F(2,6) = 48.12, Tukey's multiple comparison test, $P_{con\ versus\ LPC} = 0.0075$, $P_{LPC\ versus\ LPC+ISP} = 0.0056$). **c** Western blot analysis of MMP-2 expression in spinal cord tissue of naive, vehicle, or ISP-treated mice at 7 dpl. Data normalized to β-actin protein expression ($n = 3$ mice/group, ANOVA F(2,6) = 33.14; Tukey's multiple comparison test, $P_{con\ versus\ LPC} = 0.0168$, $P_{LPC\ versus\ LPC+ISP} = 0.0143$). **d** Representative immunohistochemistry images show Olig2[+] OPCs expressing MMP-2 in the spinal cord of ISP-treated mice at 14 dpl. White arrows indicate the colocalization of MMP-2 and Olig2. **e** Representative immunohistochemistry images show Iba1[+] cells (microglia/macrophage) expressing MMP-2 in the spinal cord of ISP-treated mice at 14 dpl. White arrows indicate the colocalization of MMP-2 and Iba1. Scare bar = 100 μm. **f** Representative eriochrome cyanine (myelin) staining of LPC lesions from the spinal cords of naive, vehicle, ISP, MMP-2 inhibitor (OA-Hy), or MMP-2 shRNA-treated mice. Dashed lines demarcate lesion areas. Scale bar = 100 μm. **g** Quantitative analysis of the volume of lesioned spinal cord in vehicle, ISP, MMP-2 inhibitor (OA-Hy), or MMP-2 shRNA-treated mice at 18 dpl ($n = 4$ mice/group, ANOVA F(5,18) = 169.7; Tukey's multiple comparison test, $P_{LPC\ versus\ LPC+ISP} < 0.0001$; $P_{LPC+ISP\ versus\ LPC+ISP+OA-Hy} < 0.0001$; $P_{LPC+ISP\ versus\ LPC+ISP+MMP-2\ shRNA} < 0.0001$; $P_{LPC\ versus\ LPC+OA-Hy} = 0.0279$). The data are presented as mean ± s.e.m. *$P < 0.05$, ***$p < 0.001$

## Methods

**Animals**. All animal care and animal procedures were approved by the Institutional Animal Care and Use Committee of Case Western Reserve University School of Medicine. Wild-type C57BL6/J mice were purchased from the Jackson Laboratory (Stock No. 000664) and housed in the Animal Research Center of Case Western Reserve University. Mice were maintained with a 12-h light/dark cycle. Both male and female mice were included in this study.

**Experimental autoimmune encephalomyelitis (EAE) model**. For induction of EAE, C57BL6/J female mice at 10-weeks of age were immunized with MOG$_{35-55}$ together with complete Freund's adjuvant emulsion (Hooke Laboratories, MOG$_{35-55}$ EAE Induction kit, EK-2110) according to the manufacturer's instructions. Use of the EAE Induction Kit resulted in 98% successful disease induction. All EAE animals were monitored daily and scored using a clinical scale from 0 to 5 (0: no abnormality; 1: limp tail; 2: limp tail and hind leg weakness; 3: limp tail and complete paralysis of hind legs; 4, hind leg and partial front leg paralysis; 5: moribund). Once EAE mice scored 1 or 3, they were randomly recruited into treatment and vehicle groups. For the treatment group, 5–7 mice were given daily intraperitoneal injections of ISP (20 µg/day), or 5% DMSO in saline, 100 µl) for Vehicle group. Experiments were blinded and animals were scored daily. For ISP Onset: ISP treatment was given at onset of sickness scored by clinical scale. For ISP Peak: ISP treatment was given at peak of sickness scored by clinical scale.

**LPC-induced focal demyelination in mice**. 12-week-old C57BL6/J male mice were anaesthetized using isoflurane and a laminectomy was performed. A volume of 1.5 µl of 1% LPC was infused into the dorsal column between T11 and T12 of the spinal cord at a rate of 0.25 µl/min. The needle was removed after a delay of 5 min to minimize back flow and the lesion closed. Starting at 24 h post-surgery, mice were treated daily with ISP (20 µg/day) or vehicle (5% DMSO in saline, 100 µl) by subcutaneous injection near the injury site. The mice were euthanized at days 7, 14, and 21. Spinal cords were dissected for further western blot, histology, and ultrastructural analyse. Control animals received an equivalent injection of saline, and tissue was collected. For the second injection of the MMP-2 inhibitor OA-Hy (10 µg/1.5 µl, 444244, Calbiochem), lentiviral particles expressing shRNA targeting mouse MMP-2 (1 µl, LPP-MSH027657-LVRU6GP, GeneCopoeia) or 0.9% saline, animals were anesthetized at 1 d (for lentiviral particles) or 4 d (for OA-Hy) after LPC lesion and OA-Hy or lentiviral particles were delivered to the same area using the above paradigm. Animals were allowed to recover and were sacrificed at 14 dpl or 18 dpl. Lesion sizes were determined by staining of serial sections with eriochrome cyanine.

**LPC-induced demyelination in mouse cerebellar slice cultures**. Cerebellar slice cultures were performed in cell culture inserts (0.4 µm, Millicell-CM, Millipore)[33]. Briefly, 300 µm-thick cerebellar slices were cut from P8-12 mouse cerebellum using a Leica vibrating microtome (Leica, VT1000S) and cultured in medium containing 50% basal eagle medium, 25% Heat-inactivated horse serum, 25% Hank's solution, 2.5% glucose, and 1% glutamine and penicillin-streptomycin. After 4 days in vitro (div), 0.5 mg/ml LPC was added for 17–18 h to induce demyelination. Slices were then incubated with 2.5 µM ISP for 8 days. For GM6001 experiments, 25 µM GM6001 (2903, Tocris), 100 nM MMP-2 inhibitor OA-Hy (444244, Calbiochem), and/or 2.5 µM ISP or SISP were incubated for 9 days. Remyelination was examined by semi-quantitative western blot of MBP and immunofluorescence staining of MBP and NF200.

**Immunostaining of cultured slices**. Slices were fixed with 4% PFA, delipidated, and washed three times in PBS, blocked in PBS containing 0.1% Triton X-100 and 5% normal goat serum and incubated with anti-MBP (SMI-99P, Covance, 1:300), and anti-NF200 (N4142, Sigma, 1:250) antibodies overnight at 4 °C. Slices were then washed in PBS and incubated in Alexa Fluor-conjugated secondary antibodies (1:500, Invitrogen) for 2 h. Slices were mounted in Vectashield mounting medium with DAPI (Vector Laboratories) and analyzed using a Leica DFC500 fluorescence microscope.

**Purified OPC cultures**. OPCs were prepared from newborn C57BL6/J mice[50]. Cell culture plates were precoated with IgM (10 µg/ml, Millipore) in 50 mM Tris-HCl followed by the addition of the primary mouse antibody A2B5. Dissociated cells were incubated in the precoated culture dishes for 30 min at 37 °C and then non-adherent cells were gently removed. A2B5$^+$ OPCs were released by 0.05% trypsin in DMEM at a purity of ~95% (Supplementary Fig. 7A). Purified OPC cells were expanded in DMEM/F12 medium supplemented with N2, 20 ng/ml PDGF, 20 ng/mlFGF, 5 ng/mlNT-3, 10 ng/ml CNTF, Glutamine (200 mM).

**CM protease activity assay**. Around $1 \times 10^6$ OPCs were plated per well on PLL, 1 µg/mL laminin, and 2 µg/mL aggrecan coated 6-well plates and treated with vehicle, 2.5 µM ISP, or SISP for 2 div at 37 °C. CM was collected, cell-strained, concentrated (Millipore Ultracel YM-3), and placed on ice until assayed. A ThermoFisher Protease Assay Kit (E66383, EnzChek, ThermoFisher) was used to assay protease activity. 1× of the EnzChek mixture was mixed 1:1 with the CM

from each group and incubated at room temperature overnight with gentle shaking. Three replicates were performed for each sample. Samples were analyzed using a spectrophotometer at 502/528 nm excitation and emission to assess cleaved and fluorescing casein. Fluorescence units reported were blanked with EnzChek and cell culture media mixture.

**CSPG gradient crossing assay and gradient quantification**. CSPG gradients were prepared on coverslips[34]. Briefly, 24-well glass coverslips were coated with poly-L-lysine and nitrocellulose, and a mixture of 700 µg/mL aggrecan (A1960, Sigma) and 10 µg/mL laminin (11243217001, Sigma) spotted on the coated coverslip. After drying, coated coverslips were then incubated with laminin at 37 °C for 3 h. Purified OPCs were plated at a density of 10,000/coverslip and cultured in DMEM/F12 medium containing with N2, PDGFRα (20 ng/ml), FGF (20 ng/ml), NT-3 (5 ng/ml), CNTF (10 ng/ml), Glutamine (200 mM). Coverslips were stained with CS56 (C8035, Sigma, 1:500), and O4 antibodies (Hybridoma Core Cleveland Clinic, 1:10). O4-positive cells crossing the aggrecan border were counted for each spot. For CSPG gradient quantification, $1 \times 10^6$ OPCs were plated per well on PLL, 1 µg/mL laminin, and 2 µg/mL aggrecan coated 6-well plates. OPCs were treated with vehicle, 25 µM GM6001 (2983, Tocris), 100 nM MMP-2 inhibitor OA-Hy (444244, Calbiochem), 2.5 µM ISP, or 2.5 µM SISP for 2 div at 37 °C. CM was harvested and incubated with freshly made spots. Spots were incubated with CM for 2 div at 37 °C then stained with CS56 and laminin (L9393, Sigma, 1:1000) antibodies and consistently imaged. Using ImageJ software (NIH), pixel intensities of the CS56 or laminin spot rims were quantified using the same region of interest saved in ImageJ.

For spinal cord explants cultured on spots, the spinal cords at cervical and thoracic levels of P1 mouse pups were chopped into 1–2 mm tissue pieces and transferred to the coverslips. Explants were cultured in DMEM/F12 medium with 15% FBS (Hyclone), 10 ng/ml PDGF (Sigma), and N2 supplement (Invitrogen). For ISP treatment, 2.5 µM ISP was added to the media at the time of plating. 25 µM GM6001 (2983, Tocris), 100 nM MMP-2 inhibitor (444244, Calbiochem), and/or 2.5 µM ISP or SISP were used.

**Peptide sequences**. Peptides were purchased from GenScript or CS-Bio in 1 mg lyophilized quantities (>98% purity) that were diluted to 2.5 mM in dH20 and aliquoted at −20 °C until ready for use[25].

ISP:GRKKRRQRRRCDMAEHMERLKANDSLKLSQEYESI
Scrambled ISP (SISP):
GRKKRRQRRRCIREDDSLMLYALAQEKKESNMHES

**Western blot analysis of aggrecan and laminin**. CM was harvested from $1 \times 10^6$ OPCs incubated on PLL, 1 µg/mL laminin, and 2 µg/mL aggrecan coated 6-well plates. OPCs were treated with vehicle, 2.5 µM ISP, or 2.5 µM SISP in conjunction with 10 µg/mL Exo1 (ab120292, Abcam), 25 µM GM6001 (2983, Tocris), or 100 nM MMP-2 inhibitor (444244, Calbiochem) for 2 days at 37 °C. 100 µL CM of each cell-strained group was incubated with 20 µg/mL aggrecan and/or 10 µg/mL laminin for 2 h in 1 mL Eppendorf tubes at 37 °C. As a positive control, OPC media was incubated with aggrecan and incubated in the same fashion. Western blots were then performed as described below with incubation against CS56 and/or laminin antibodies.

**Western blot analysis of OPC lysate or CM**. To assess MMP-2 or 10 in OPC CM or lysates, OPCs were incubated on precoated PLL, aggrecan, and laminin as described above. OPCs were treated with vehicle, 2.5 µM ISP or SISP for 2 days at 37 °C. CM was then harvested and concentrated using Millipore Ultracel YM-C centrifugal filter units at max speed (Eppendorf Centrifuge 5415D) for 30 min at 4 °C. A volume of 50 µg of concentrated CM was loaded into each lane. OPC lysates were assessed using western blot techniques described below with 20 µg protein loaded from each group.

**Western blot analysis**. Tissue samples or cerebellar slices or purified OPC cells were homogenized with RIPA lysis buffer and protein concentration was determined by Pierce BCA protein assay kit according to the manufacture's instructions (Thermo Fisher). Then, equal amounts of protein were loaded onto 15% SDS-PAGE gels, and electrophoretically transferred to PVDF membranes (Millipore). The membranes were blocked in 0.1% TPBS buffer with 5% non-fat milk for 1 h at room temperature and probed with indicated primary antibodies overnight at 4 °C and followed by secondary antibodies conjugated to horseradish peroxidase (HRP). The following primary antibodies were used: MBP (SMI-99P, Covance, 1:1000), CS56 (C8035, Sigma, 1:1000), Laminin (L9393, Sigma, 1:1000), GAPDH (AF5718, R and D Systems, 1:1000), MMP-2 (AF1488, R and D Systems, 1:1000), MMP-10 (MAB910, R and D Systems, 1:1000), and β-actin (sc-47778, Santa Cruz, 1:1000). Enhanced chemiluminescence was performed with a West Pico Kit (Thermo Fisher) and detected by FluorChem E system (ProteinSimple, USA). The density of bands was quantified using ImageJ software (NIH). Image has been cropped for presentation (full-size image is shown in Supplementary Figs. 12, 13).

**Gelatin zymography**. CM was collected from $1 \times 10^6$ OPCs incubated on PLL, 1 µg/mL laminin, and 2 µg/mL aggrecan coated 6-well plates. OPCs were treated with vehicle, 2.5 µM ISP or SISP for 2 days at 37 °C. CM was concentrated using Millipore Ultracel YM-3 centrifugal filter units as previously described. A volume of 40 µg undenatured protein from concentrated CM or 25 ng of activated recombinant MMP-2 (PF023, Millipore) and MMP-9 (PF140, Millipore) was loaded with 1× laemmli buffer (1610747, Bio-Rad) onto 10% gelatin zymograms (1611167, Bio-Rad) and run at 100 mV for 1.5 h on ice. Zymograms were then gently shaken with 1× renaturing buffer (1610765, Bio-Rad) for 30 min at RT then incubated with 1× developing buffer (1610766, Bio-Rad) O/N at 37 °C. Developed zymograms were then gently washed with dH20 and incubated O/N with 0.1% Coomassie Blue dye (27816, Sigma). Following washes with destaining buffer (40% MeOH, 10% Acetic Acid), zymograms were imaged and the amount of gelatin degradation was assessed using ImageJ via a method described in the western blot section.

**Protease array screen**. OPCs were cultured on 6-well plates for 4 div with vehicle control or 2.5 µM ISP. CM was collected from each group and cell-strained before incubation with a blot array provided in the R & D Systems Protease Array Kit (ARY025). Instructions with the kit were followed using overnight incubation of collected media at 4 °C. Control and ISP blots were developed together with the same exposure time and pixel intensities of the array were assessed using ImageJ (NIH).

**LFB myelin staining and quantification**. LFB staining was performed according to the manufacturer's instructions (#26681, Electron Microscopy Sciences). For spinal cord sections, 20 µm coronal sections were incubated in LFB solution in 56 °C overnight and then rinsed sequentially with 95% alcohol and distilled water. The sections were then placed in 0.1% lithium carbonate solution and dehydrated with a series of graduated ethanol, cleared with Histoclear, and mounted. A set of serial matched sections were imaged and analyzed. Images (5 to 6 sections/animal) were captured under the light microscope. The demyelinated areas (lack of LFB staining) were quantified using ImageJ software. For EAE, demyelinated areas were measured and represented as a percentage of total area of spinal cord. For sections of the LPC model, lesion volumes were calculated by the lesion area from serial sections throughout the entire lesion based on the equation for volume of a cylinder ($V =$ lesion area × length of lesion).

**Immunocytochemistry**. For MMP-2/O4 staining, OPCs were plated onto 24-well coverslips that were precoated with PLL, 1 µg/mL laminin, and 2 µg/mL aggrecan and incubated with vehicle or 2.5 µM ISP for 2 days at 37 °C. Cultured OPC or OL cells were fixed in 4% PFA and followed by blocking in PBST solution (10% normal goat serum and 0.2% Triton–X100 in PBS). Diluted primary antibodies were incubated with samples overnight at 4 °C and followed by appropriate secondary antibody goat anti-mouse or anti-rabbit IgM or IgG conjugated with Alexa Fluor 488 or 594 (1:500, Invitrogen). The following primary antibodies were used: PDGFRα (ab65258, Abcam, 1:250), O4 (Hybridoma Core, Cleveland Clinic), MBP (SMI-99P, Covance, 1:300), MMP-2 (Ab19167, Millipore, 1:200), NG2 (AB5320, Millipore, 1:250), and CS56 (C8035, Sigma, 1:250). Cells were mounted with Vecta Shield mounting medium with DAPI (Vector Laboratories).

**TUNEL, Ki67 immunocytochemistry and quantification**. To assess proliferation, OPCs were plated on 24-well coverslips that were precoated with PLL, 1 µg/mL laminin, and 2 µg/mL aggrecan. OPCs were treated with vehicle, 100 nM MMP-2 inhibitor (444244, Calbiochem), and/or 2.5 µM ISP or SISP immediately upon plating for 2 days at 37 °C. Coverslips were fixed and stained using the same method described in the immunocytochemistry section with Ki67 (550609, BD Pharmingen, 1:500) and O4 (1:10). Coverslips were imaged and counted. To assess apoptosis, OPCs were cultured in the same fashion and incubated with vehicle or 1 µg/mL LPC for 2 h at 2 days following vehicle, 2.5 µM ISP or SISP incubation. Coverslips were then fixed with PFA and methanol, then stained using an APO-BrdU TUNEL assay kit (A23210, ThermoFisher). We followed the manufacturer's guidelines with an overnight incubation of the primary antibody at room temperature. Coverslips were co-stained with DAPI (D9542, Sigma, 1:10,000) then imaged and counted.

**Immunohistochemistry**. Mice were anesthetized with avertin and perfused with PBS and 4% paraformaldehyde (PFA). Spinal cords were dissected and post-fixed in 4% PFA overnight at 4 °C and equilibrated in 20% sucrose. 20 µm-thick sections were pretreated with Reveal Decloaker Solution (RV1000M, Biocare Medical) for antigen retrieval according to the manufacturer's instructions. After blocking, sections were incubated with primary antibodies overnight at 4 °C and followed by appropriate secondary antibodies conjugated with Alexa fluorescence 488 or 594. The following primary antibodies were used: MBP (SMI-99P, Covance, 1:300), NeuF200 (N4142, Sigma, 1:250), CS56 (C8035, Sigma, 1:250), MMP-2 (AB191677, Sigma, 1:500), laminin (L9393, Sigma, 1:500), CAT301 (MAB5284, Millipore, 1:250), versican (AB1032, Millipore, 1:250), Iba1 (019-19741, WAKO, 1:250), GFAP (MAB360, Millipore, 1:250), Olig2 (AB9610, Millipore, 1:250), CC1 (OP80, Millipore, 1:250), iNOS (#610329, BD Biosciences, 1:250), Arginase-1 (sc-18355, Santa Cruz, 1:50), PDGFRα (Cell signaling, #3174), and NG2 (Millipore, AB5320). For each staining, at least three individual animals/group were examined and

images were captured with a Leica DFC500 fluorescence microscope. Staining was quantified using Image J software (US National Institutes of Health, USA). Fluorescence intensity was calculated as percentages of the mean value of the naive controls.

**Tissue preparation for electron microscopy (EM) analysis**. For ultrastructural analyses of myelination, anesthetized animals were perfused with 2% glutaraldehyde/ 4% PFA in 0.1 M sodium carcodylate buffer, pH 7.4 (Electron Microscopy Sciences). Lesioned areas of the LPC or EAE-induced spinal cords from ISP-treated or control animals were dissected and post-fixed in 1% OsO4 for 2hs. Coronal sections (500 µm) of spinal cord were prepared (Leica, Vibratome), dehydrated, stained with saturated uranyl acetate and embedded in a Poly/Bed812 resin (Polysciences Inc.). 1 µm-thick sections were cut and stained with toluidine blue, and matched areas were selected for EM analysis. For ultrastructure analysis, ultrathin sections (0.1 µm) were cut and visualized using an electron microscope (JEOL100CX) at 80 kV. G-ratios were calculated from 50–100 randomly selected myelinated axons by measuring the myelin thickness of the inner and outer diameters of the myelin sheath.

**shRNA knockdown of MMP-2**. MMP-2 knockdown was mediated through lentiviral particles expressing shRNA targeting mouse MMP2 driven by the U6 promoter (LPP-MSH027657-LVRU6GP, GeneCopoeia). Lentiviral particles for shRNA scrambled constructs (LPP-CSHCTR001-LVRU6GP-100-C, GeneCopoeia) were used as the corresponding controls. OPC cultures were infected for at least 48 h before experiments at the multiplicity of infection (MOI) of 1. Constructs were validated using western blot analysis of infected OPC cultures for MMP-2 (AB191677, Sigma, 1:500).

**Statistical analyses**. All data analyses were performed using GraphPad Prism 6.00. Data are shown as mean ± SEM. $p < 0.05$ is deemed statistically significant. Statistical analysis was performed by two-tailed unpaired Student's $t$ tests, one-way or one-way ANOVA with posthoc analysis by Tukey's multiple comparison test, Dunnett's multiple comparison test, or Sidak's multiple comparison test. Quantifications were performed in a blinded fashion. No statistical tests were used to predetermine sample sizes, but our sample sizes are similar to those generally employed in the field. Data distribution was assumed to be normal, but this was not formally tested. All experiments were performed at least three times independently.

## Data availability

The data that support the finding of this study are available from the corresponding author upon reasonable request.

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

## Acknowledgements

This work was supported by the National Institutes of Health (R01 NS077942 to Y.Y.; NS025713 to J.S.).

## Author contributions

F.L., A.P.T, Y.Y., and J.S designed research and interpreted data; F.L. and A.P.T performed experiments with assistance from L.X. and C.S.; L.X. performed the LPC surgery; and C.S. performed microscopy, data gathering, and imaging. B.T.L provided ISP and contributed manuscript editing; F.L., A.P.T, Y.Y., and J.S wrote the manuscript.
