## [Peer Review File · Nature Communications]

Reviewers' comments:

Reviewer #1 (Remarks to the Author):

In this paper by Yang and colleagues, the authors explore the role of PTP inhibition in mouse models of MS and remyelination. Previously this group generated a peptide inhibitor of PTP, called ISP that can block CSPG activity and aid spinal cord injury. In this study, they seek to apply this ISP approach to EAE and LPC models of white matter damage. There are three parts to this paper. The first part of the paper shows very clearly that ISP treatment suppresses EAE and enhances remyelination after LPC. The second part shows that ISP appears to promote OPC migration in vitro into areas that have CSPG barriers. In the third part, they seek to address how OPC migration is enhanced and show that ISP promotes MMP2 activity in OPCs and that MMP2 activity is important for remyelination. In general the first part of the paper (figure 1-4) are well done and very nicely show that ISP can attenuate the effects of EAE and LPC. The mechanistic parts of the paper, while very interestingly are lacking in vivo relevance and need further refinement before this paper is suitable for publication. Below are a series of suggestions aimed at improving this paper:

1) In Fig4A the Iba marker is drastically reduced. They need to stain for a series of microglia and other immune markers, and also reactive astrocytes in their EAE experiments. While the data showing that these lineages are not affected by ISP is clear for LPC, it looks like the inflammatory response is affected by ISP in EAE.

2) In Figure 5, the PDGFR cells that have "crossed" the barrier do not at all look like OPCs. For that matter the cells that remain on the other side also do resemble OPCs. This leads to two questions: 1) Does crossing the barrier provoke differentiation? and 2) Are those really OPCs? Both can be resolved with additional marker staining.

3) Along these same lines, what is the evidence that ISP promotes migration of OPCs into the actual lesion in vivo? They show data in supplement, in LPC model that OPC proliferation is unaffected. Can they also show that over time there is an accumulation of OPCs in the lesion in the presence of ISP? I would imagine that over time the number of OPCs occupying the lesion in the presence of ISP will increase compared to control.

4) The logic of how they stumbled upon MMP2 is not really made clear.

5) The model is that OPC-derived MMP2 activity is enhanced by ISP and this mediates the remyelination effects. If so the authors have not sufficiently proven that this mechanism is relevant to the in vivo conditions they are modeling and need to address the following issues:

a) What happens in EAE? EAE is not a remyelinating model, so the OPC-MMP2 model is not applicable.

b) Is MMP2 expressed in OPCs in human MS lesions, OPCs in mouse brain/spinal cord in normal and injury conditions? Is its expression increased after injury?

c) Are the effects of ISP on OPC migration the result of MMP2 activity? For this experiment they turned to pharmacology and have some evidence that it may. However this is not sufficient evidence for this reviewer. To prove this, they need a genetic manipulation: delete MMP2 and ask whether the effects of ISP are attenuated---or overexpress activated MMP2 and ask if it mimics ISP. Either of these manipulations can be achieved in vitro and in the in vivo LPC model.

Reviewer #2 (Remarks to the Author):

Little is known about the role of extracellular matrix components and proteases in remyelination after myelin injury in either inflammatory or non-inflammatory models. This paper now provides new insights into this process. The authors report an upregulation of CSPG in lysolecithin-induced or EAE lesions after demyelination. They provide evidence that CSPGs binds to PTP receptor on OPCs, which regulate the secretion of metalloproteases. Metalloproteases in turn enzymatically digest the CSPGs to promote OPC migration into lesions. Furthermore, they report that a small peptide is able to modulate this pathway in such a way that remyelination is enhanced. These findings are potentially important and of great interest, but the quality of the study needs to be improved. Many of the experiments are not properly quantified and controlled.

- The authors state on page 5 that the CSPG stainings are increased, but this is not quantified in Supp Fig1
- A major concern is the interpretation of the EAE experiments on page 7. The authors report differences in the EAE score by peptide treatment and show differences in myelin content in Fig 1. They conclude this is due to differences in remyelination, but it is not clear whether the EAE score changes because of effects on inflammation or remyelination. Likewise it is not clear whether the number of myelinated axons change because of differences in demyelination or remyelination. The authors need to address these points for example by using specific markers to determine the number of remyelinating oligodendrocytes.
- All of the blots showing MBP have an unusual pattern of the bands. The full blots are cut in such a way that the 21kDa band that appears above the 14 and 18 kDa cannot be seen. The authors should provide full scans and use a second MBP antibody to control these experiments.
- It is not clear how the EM experiments in Figure 1 were quantified. How do the authors recognize unmyelinated axons in the lesions? Most axons are damaged and it is not easy recognize all axons by EM. The authors should report the data as number of myelinated axons instead of % myelinated axons.
- In Figure 2E the number of myelinated axons is very low in control at 14 dpi - only 20%. The authors should re-quantify.
- The staining shown for PTP and Olig2 in SuppFig2 is not convincing and not quantified. If available, the authors need to use a second antibody to confirm or provide evidence that

staining is specific.

- On page 11 the authors write that CSPG degradation is much slower ... (Supp Fig 3), but the data is not quantified
- Page 16 and Figure 6I/J the authors report finding using Exo1, but is unclear how this inhibitor blocks exocytosis. This is a very crude approach that needs to be confirmed using an independent assay
- The data on MMP2 is exclusively based on pharmacological inhibitors, which could be unspecific. Genetic experiments for example using shRNA are needed for this conclusion.

Address of Reviewer #1's Comments:

1.1 In Fig4A the Iba marker is drastically reduced. They need to stain for a series of microglia and other immune markers, and also reactive astrocytes in their EAE experiments. While the data showing that these lineages are not affected by ISP is clear for LPC, it looks like the inflammatory response is affected by ISP in EAE.

The reviewer has pointed out that the Iba marker for macrophages is reduced following ISP treatment. Per the reviewer's suggestion to "stain for a series of microglia and other immune markers, and also reactive astrocytes," we performed additional immunohistochemistry of spinal cord sections from control vs. ISP treatments of EAE-induced animals.

In comparing vehicle control and ISP in Fig.4A and Sup.Fig.3A-B, the macrophage/microglia marker, Iba1 is, indeed, reduced following ISP treatment. In both groups, Iba1 still colocalizes with the chondroitin sulfate proteoglycan, aggrecan (Cat301) suggesting probable direct macrophage/microglia-to-aggrecan interactions possibly through phagocytosis. As suggested by the reviewer, we also immunostained for GFAP and saw a decrease of reactive astrocytes following ISP treatment (Sup.Fig.3A-B), which may further suggest that ISP treatment modulates the inflammatory environment to decrease astrocyte reactivity following EAE induction.

A recent study by Dyck et al. (<https://www.ncbi.nlm.nih.gov/pubmed/29558941>) has additionally addressed the reviewer's comment that Iba1 is decreased following ISP treatment. Using peptides, including ISP, to modulate the LAR family of receptors, Dyck et al. found that ISP modulated RPTP σ in such a way as to reduce markers for M1-like inflammation and promote M2 markers in microglia/macrophages following spinal cord injury. The additional effects of ISP in promoting M2-like inflammation may be one other way we are seeing remyelination effects following demyelinating injuries. To begin verifying these results in our model of ISP treatment following EAE, we immunostained vehicle and ISP treated groups against a commonly utilized M1 macrophage/microglia marker, iNOS and found that ISP treatment decreased its expression as quantified by fluorescence intensity (Sup.Fig.3C-D). Furthermore, the M2 polarization marker, Arginase-1 seems to be increased following ISP treatment compared to vehicle control. Together, these results suggest that ISP modulation of RPTP σ may additionally improve function and remyelination by modulating macrophage/microglia towards an M2 polarization. The results and discussion has been amended to reflect these recent findings.

1.2 In Figure 5, the PDGFR cells that have "crossed" the barrier do not at all look like OPCs. For that matter the cells that remain on the other side also do resemble OPCs. This leads to two questions: 1) Does crossing the barrier provoke differentiation? and 2) Are those really OPCs'? Both can be resolved with additional marker staining.

The reviewer has expressed concerns that the PDGFR α^+ cells crossing the aggrecan barrier "do not at all look like OPCs." We would like to take the opportunity to point out that the immunopanning protocol used to harvest OPCs have been previously reported to yield cultures of OPCs with around 96% purity (<http://www.jneurosci.org/content/36/10/3024.long#sec-2>).

To additionally address the two questions: 1) if CSPGs provide differentiation of OPCs and 2) if they are really OPCs, we would like to highlight Fig. 5B where the same experiment was performed with O4 antibody staining. Unlike PDGFR α , O4 has been widely used as an antibody specific to oligodendrocytes (<https://www.sciencedirect.com/science/article/pii/S0012160681904772?via%3DIihub>). Even with O4 staining, the same amount of OPCs, as seen in Fig. 5A with PDGFR α immunostaining, are able to cross the aggrecan barrier.

1.3 Along these same lines, what is the evidence that ISP promotes migration of OPCs into the actual lesion in vivo? They show data in supplement, in LPC model that OPC proliferation is unaffected. Can they also show that over time there is an accumulation of OPCs in the lesion in the presence of ISP? I would imagine that over time the number of OPCs occupying the lesion in the presence of ISP will increase compared to control.

The reviewer has asked for “the evidence that ISP promotes migration of OPCs into the actual lesion *in vivo*.” As the reviewer had pointed out, we had performed immunohistochemistry analysis of OPCs with Ki67, a marker of proliferation, in LPC-demyelinated spinal cords and showed that ISP did not induce increased proliferation of this cell type in the control (Sup. Fig.5). To provide more direct evidence that OPC migration is enhanced by ISP treatment through MMP-2, we performed LPC injections in conjunction with ISP with or without an MMP-2 inhibitor (Sup. Fig. 11). As anticipated, the amount of Olig2 positive cells increased in the ISP treatment group while the addition of the MMP-2 inhibitor decreased the accumulation of these cells in the LPC lesion epicenter (Sup. Fig. 11E). This correlated with a concomitant decrease in CS56 immunoreactivity (Sup. Fig. 11C).

1.4 The logic of how they stumbled upon MMP2 is not really made clear.

As “the logic of how [we came across] MMP-2 is not really made clear,” we have amended the text for clarity to better explicate how MMP-2 was selected from the screen. To reiterate here, MMP-2 was one of the many proteases included in a broad screen including different families of proteases. This screen was performed with ISP-treated conditioned media. Highly enriched proteases of each family found through the screen were then validated through western blotting of ISP-treated conditioned media and only MMP-2 was confirmed to be more significantly up-regulated in ISP-treated conditioned media. Interestingly, MMP-2 has also been cited in the regeneration literature for the ability to enhance neurite growth by degrading chondroitin sulfate proteoglycans while sparing laminin (<http://www.jneurosci.org/content/18/14/5203.long>) much as we have observed following laminin and CS56 western blotting. Furthermore, MMP-2 has been implicated in myelination peripherally (<http://onlinelibrary.wiley.com/doi/10.1002/glia.20774/abstract>).

1.5 The model is that OPC-derived MMP2 activity is enhanced by ISP and this mediates the remyelination effects. If so the authors have not sufficiently proven that this mechanism is relevant to the in vivo conditions they are modeling and need to address the following issues.

The reviewer has concerns that the model of OPC-derived MMP-2 activity enhanced by ISP to induce remyelination was not sufficiently proven to be relevant *in vivo*. We have provided additional context as well as additional *in vivo* experimentation of our hypothesis that increased MMP-2 activity occurs following ISP modulation of RPTP σ to enhance remyelination following LPC-induced demyelination. Detailed explication is provided below:

1.5a What happens in EAE? EAE is not a remyelinating model, so the OPC-MMP2 model is not applicable.

The reviewer has expressed concerns over “what happens in EAE” since, in their opinion, “EAE is not a remyelination model of MS”. To begin, while remyelination does not usually occur in this model without treatment, it has been detectable ultrastructurally through G-ratio quantification following electronmicroscopy. This has been reiterated by Jones et al. 2008 (<https://www.sciencedirect.com/science/article/pii/S0165572808001732?via%3Dihub>) who performed exhaustive histological characterization of EAE, MOG-immunized mice. Additional characterization of axon injury in lesions by Kornek et al. 2000 found remyelinated shadow plaques in rat EAE models (<https://www.sciencedirect.com/science/article/pii/S0002944010645373>). Kornek et al. further defined remyelinated shadow plaques as lesions with thin myelin sheaths, but with the pronounced presence of PLP mRNA in oligodendrocytes.

Furthermore, many reviews (<https://www.ncbi.nlm.nih.gov/pmc/articles/PMC3229753/>) have indicated that the EAE model is indeed used by some groups as a model for understanding remyelination *in vivo*. A Pubmed search for “remyelination EAE” in the title article yielded over 200 peer-reviewed articles as of March 2018. Some notable examples whereby EAE was shown to have improvements in remyelination include works by Yang et al. where adult neural stem cells specifically expressing IL-10 transplanted in a model of EAE yielded enhanced G-ratios similarly determined by electron microscopy (<https://www.ncbi.nlm.nih.gov/pmc/articles/PMC2786785/>), and Brambilla et al. (<https://www.ncbi.nlm.nih.gov/pubmed/24357067>) where remyelinating axons also identified through electron microscopy were found to be increased in a mutant where pro-inflammatory factor, I κ B α , was specifically knocked out in astrocytes compared to wild type control following EAE induction.

In our own study, we have sought to characterize remyelination in EAE beyond luxol fast blue staining to include MBP immunoblotting, and importantly, G-ratio comparisons following electron microscopy. As the reviewer has pointed out, the OPC-MMP-2 model may not be the only mechanism by which ISP is inducing remyelination in EAE. As we have addressed in point 1.1, a recently published study by Dyck et al. 2018 points to the ability of ISP modulation of RPTP σ to skew immune cells toward an M2-like phenotype to effectively reverse the M1-like phenotype induced by chondroitin sulfate proteoglycans. We maintain that the OPC-MMP-2 mediated degradation of chondroitin sulfate proteoglycans may additionally aid in either a M2-like shift in this regard or, at the least, relieve OPCs of CSPG-induced inhibition.

1.5b Is MMP2 expressed in OPCs in human MS lesions, OPCs in mouse brain/spinal cord in normal and injury conditions? Is its expression increased after injury?

To address the reviewer's question as to whether MMP-2 is expressed in human MS lesions and human OPCs, we have found the following supporting literature:

There have been reported studies where MMP-2 has been observed to be expressed in human Olig2+ glioblastoma (<https://www.ncbi.nlm.nih.gov/pubmed/17951409>, <https://www.ncbi.nlm.nih.gov/pubmed/14666700>). MMP-2 is expressed in other progenitor-type cells such as human oligodendrocytes generated from human neural stem cells: (<http://onlinelibrary.wiley.com/doi/10.1002/term.2042/full>). Moreover, others have found an enrichment of MMP-2 gene transcripts in human oligodendrocytes compared to neurons and mature or immature astrocytes (http://web.stanford.edu/group/barres_lab/cgi-bin/geneSearchMariko.py?geneNameIn=MMP2). Furthermore, RNAseq has identified MMP-2 in a database collated from human FACS sorted oligodendrocytes in the Barres Lab ([http://www.cell.com/neuron/fulltext/S0896-6273\(15\)01019-3](http://www.cell.com/neuron/fulltext/S0896-6273(15)01019-3)).

Regarding the presence of MMP-2 in human MS cases, minimal MMP-2 induction was found in human multiple sclerosis lesions and plaques although they seem to be restricted to "early" plaques (<https://academic.oup.com/brain/article/124/9/1743/303208>). Other MMPs are also upregulated in human multiple sclerosis lesions, however, as the up-regulation of other MMPs but not MMP-2 has been identified in the sera of multiple sclerosis patients (<https://www.ncbi.nlm.nih.gov/pubmed/27555667>, <https://www.ncbi.nlm.nih.gov/pubmed/24516744>).

Additionally, we have performed experiments that we hope will fully address the reviewer's question whether MMP-2's expression is increased after injury. Following immunohistochemistry of the dorsal columns of non-injured spinal cords (Fig. 8A), MMP-2 expression seems to be found predominantly in the grey matter with low baseline expression in the white matter. We found that at 14 days following LPC injection, the demyelinated dorsal column of the spinal cord exhibits some increased MMP-2 expression. Following ISP treatment of the LPC-injected dorsal column, MMP-2 expression is significantly increased over LPC injected control. While previously reported studies have highlighted the deleterious role of nonspecific protease dysregulation following pro-inflammatory environments induced by MS pathology, we propose here that controlled punctate release of MMP-2, perhaps in an M2-modulated context, provides CSPG-degrading effects that allows for myelin regeneration following demyelination injury.

1.5c Are the effects of ISP on OPC migration the result of MMP2 activity? For this experiment they turned to pharmacology and have some evidence that it may. However this is not sufficient evidence for this reviewer. To prove this, they need a genetic manipulation: delete MMP2 and ask whether the effects of ISP are attenuated-or overexpress activated MMP2 and ask if it mimics ISP. Either of these manipulations can be achieved in vitro and in the in vivo LPC model.

To address the reviewer's concern with pharmacological inhibition of MMP-2, we have performed additional experiments involving the shRNA-mediated knock down of MMP-2 delivered to OPCs through lentiviral particles. Through western blotting for MMP-2, we have

verified that our construct indeed knocks down MMP-2 compared to scrambled lentiviral control. We have applied these constructs in conjunction with ISP in our *in vitro* experiments and found that: 1) shRNA knockdown of MMP-2 was unable to fully rescue ISP-mediated enhancement of MBP+ area in oligodendrocytes grown on a substrate of laminin and aggrecan *in vitro*, 2) that shRNA knockdown of MMP-2 decreased ISP-mediated migration of OPCs past a barrier of highly concentrated chondroitin sulfate proteoglycans in our spot assay, and 3) that shRNA knockdown of MMP-2 decreased the ability of ISP to remyelinate LPC-demyelinated cerebellar slices *ex vivo*.

Most importantly, in addition to *in vitro* and *ex vivo* experiments, we tested the same shRNA constructs *in vivo* in an LPC model. As described in the amended text, Fig.8A-B depicts a low baseline of MMP-2 expression in the white matter of the non-injured dorsal aspect of the spinal cord. While MMP-2 is slightly up-regulated following 14 days after LPC-induced demyelination injury, ISP treatment is able to approximately double the expression of MMP-2. This is confirmed in western blot immunostaining against MMP-2 normalized by β -actin in Fig.8C. Immunohistochemistry of ISP-treated LPC lesions at 14 days post-injection depict Olig2 colocalizing with MMP-2. Finally, we tested whether MMP-2 acted through ISP treatment to affect remyelination. The same MMP-2 inhibitor resulting in decreased remyelination *ex vitro* was injected intraspinally 3 days following LPC injection at the same site. At 18 days post lesion, MMP-2 inhibition itself was able to enlarge the lesion volume of myelin significantly over LPC only control as visualized via immunostaining with eriochrome cyanine. While ISP treatment of LPC injected animals significantly reduced the lesion volume compared to LPC only control, the addition of the MMP-2 inhibitor significantly decreased this effect. Using a shRNA-mediated knock down of MMP-2, we additionally confirm that a decrease in MMP-2 is able to decrease the efficacy of ISP-mediated myelin regeneration in this model.

Together, we hope these additional experiments help to better convey that ISP-mediated regeneration of myelin is, in part, through the up-regulated expression of MMP-2 activity. This has been further validated *in vivo* using an LPC model and, with the addition of shRNA-targeting MMP-2 constructs, we hope now is highly convincing that MMP-2 activity in particular is important in ISP-induced regeneration following models of demyelinating injuries. Amendments in the text additionally reflect these changes.

Address of Reviewer #2's Comments:

2.1 The authors state on page 5 that the CSPG stainings are increased, but this is not quantified in Supp Fig1.

Per the reviewer's suggestion, we have provided quantification of the chondroitin sulfate proteoglycan staining in Supp.Fig1. In the naïve cord, immunohistochemistry staining of the chondroitin sulfate proteoglycan, aggrecan, is low especially in the white matter area. Aggrecan or CSPG sugar moieties as assessed by two different antibodies, Cat301 and CS56 respectively, greatly increases at 28 days post EAE induction. Aggrecan levels approximately double at 41 days post EAE induction. In the LPC model, however, aggrecan immunostaining increases at 7 days post lesion and gradually decreases.

2.2 A major concern is the interpretation of the EAE experiments on page 7. The authors report differences in the EAE score by peptide treatment and show differences in myelin content in Fig 1. They conclude this is due to differences in remyelination, but it is not clear whether the EAE score changes because of effects on inflammation or remyelination. Likewise it is not clear whether the number of myelinated axons change because of differences in demyelination or remyelination. The authors need to address these points for example by using specific markers to determine the number of remyelinating oligodendrocytes.

The reviewer has pointed to an important concern in the interpretation of the EAE experiments. Specifically, there may be some contention over whether ISP-induced improvements in the EAE score may be due to effects on inflammation. Indeed, a recent study from Dyck et al. 2018 (<https://www.ncbi.nlm.nih.gov/pubmed/29558941>) provides support and additional context to the reviewer's interpretation that the reduction of chondroitin sulfate proteoglycans after peptide treatment occurs, in part, through macrophage protease-led phagocytosis and degradation of the proteoglycan itself.

Indeed, in agreement with the reviewer, it is clearly possible that ISP treatment may be modulating the inflammatory environment following demyelinating injury. Dyck et al., have recently shown that ISP treatment following spinal cord injury modulates the inflammatory environment of the spinal cord to skew immune cells toward an M2-like polarization instead of an M1-like polarization. Chondroitin sulfate proteoglycans alone are able to induce different immune cells such as microglia/macrophages and T cells to exhibit more M1 markers. The addition of ISP, however, was able to increase M2-like markers in these same cells as assayed through several different techniques such as western blot, flow cytometry, and immunohistochemistry. The ability of chondroitin sulfate proteoglycans to modulate macrophages has additionally been observed by the Bradbury group who used both a lentiviral (<https://www.ncbi.nlm.nih.gov/pubmed/24695702>) and conventional one-time intraspinal delivery (<https://www.ncbi.nlm.nih.gov/pubmed/25471580>) of a chondroitin sulfate proteoglycan cleaving enzyme, chondroitinase, to show that macrophages skew toward expression of more M2 markers and anti-inflammatory factors such as IL-10. Treatment of ISP to modulate RPTP σ in our model of EAE may therefore be modulating the inflammatory environment toward an M2-like phenotype through this receptor. Importantly, we have now stressed the co-localization of aggrecan with Iba1⁺ macrophages and we have provided additional experiments using immunohistochemistry and we have compared and quantified EAE-induced vehicle vs. ISP treated groups and found that iNOS (M1) expression decreases following ISP treatment while Arginase-1 (M2) expression increases. Considerable discussion has now been added to call attention to the role of immune cell modulation by ISP in the phenomena of proteoglycan removal and remyelination.

Finally, to address the concern over whether the number of myelinated axons after peptide treatment may be changing because of differences in initial demyelination or remyelination, we have provided additional quantification of luxol fast blue stained myelin at 18 days post EAE induction (a time point in which clinical scores averaged around 3 in all groups), which show no differences between vehicle and ISP groups (Sup. Fig. 4E,F). Thus, disease onset is similar

between all three groups as we observed a delayed recovery response and no significant changes in initial demyelination.

2.3 All of the blots showing MBP have an unusual pattern of the bands. The full blots are cut in such a way that the 21kDa band that appears above the 14 and 18 kDa cannot be seen. The authors should provide full scans and use a second MBP antibody to control these experiments.

The reviewer has stated “All of the blots showing MBP have an unusual pattern of the bands.” We used the SMI-99P antibody from Covance to detect MBP. This antibody has been cited to specifically detect developing and adult myelin and developing oligodendrocytes and was developed to particularly distinguish oligodendrocytes from astrocytes, microglia, neurons, or other cells of the CNS. Specifically, the antibody recognizes the 21kDa band and does not recognize the 14kDa form. The reviewer claims that the “blots have been cut in such way that 21kDa band that appears above the 14 and 18kDa cannot be seen.” However our western blots yield approximately two bands around 21 and 18kDa consistent not only with blots we have previously published in an unrelated study (<https://www.ncbi.nlm.nih.gov/pubmed/28238799>), but independently verified with blots other groups have published using the same antibody (<https://www.ncbi.nlm.nih.gov/pubmed/27581459>, <https://www.ncbi.nlm.nih.gov/pubmed/27306410>).

2.4 It is not clear how the EM experiments in Figure 1 were quantified. How do the authors recognize unmyelinated axons in the lesions? Most axons are damaged and it is not easy recognize all axons by EM. The authors should report the data as number of myelinated axons instead of % myelinated axons.

We would like to address the reviewer’s inquiries regarding how electron microscopy experiments were quantified here as well as with corresponding changes to the text in “Methods.” To calculate the G-ratio, we analyzed the ratio of the inner diameter of the myelin sheath to its outer diameter between matching lesioned spinal cords of LPC or EAE-induced ISP-treated and control tissue blocks. At least 50-100 randomly selected myelinated axons were used to assess the G-ratio. As recommended by the reviewer, we have additionally reported the data in Fig. 2I as the number of myelinated axons per field instead of by percentage of myelinated axons.

2.5 In Figure 2E the number of myelinated axons is very low in control at 14 dpi - only 20%. The authors should re-quantify.

The reviewer states that the number of myelinated axons is low in control at 14dpi in Fig. 2E. We have expressed the data as a number of myelinated axons per field instead of as a percentage. We would also like to point out that other groups have reported control levels of myelinated axons as low as 6% following electron microscopy and G-ratio analyses around 12dpi (<https://www.nature.com/articles/nature14335>)

2.6 The staining shown for PTP and Olig2 in SuppFig2 is not convincing and not quantified. If available, the authors need to use a second antibody to confirm or provide evidence that staining is specific.

The reviewer has expressed a concern with the RPTP σ and Olig2 immunostaining in Supp. Fig. 2A. Specifically, that “the staining shown for RPTP σ and Olig2 in Supp. Fig. 2 is not convincing.” We would like to take the opportunity to clarify that this piece of RPTP σ /Oligodendrocyte immunocytochemistry was shown in conjunction with immunocytochemistry of RPTP σ with MBP and CC1 markers (Supp. Fig. 2C).

2.7 On page 11 the authors write that CSPG degradation is much slower ... (Supp Fig 3), but the data is not quantified.

The reviewer has specified “the authors write that CSPG degradation is much slower [in] Supp. Fig. 3, but the data is not quantified.” To address this concern, we have quantified CSPG degradation in our cerebellar slice cultures following ISP or vehicle treatment. Aggrecan (Cat301) expression is not significant at 4 days post lesion in control and ISP treated groups; however, by 8 and 14 days post lesion, ISP treatment significantly depletes aggrecan immunostaining.

2.8 Page 16 and Figure 6I/J the authors report finding using Exo1, but is unclear how this inhibitor blocks exocytosis. This is a very crude approach that needs to be confirmed using an independent assay.

The reviewer has commented on concerns with the use of Exo1 as an exocytosis inhibitor as it is unclear how this drug blocks exocytosis. To clarify, the small molecule Exo1 was identified as a reversible inhibitor of exocytosis by Feng et al. 2003 (<http://www.pnas.org/content/100/11/6469.long>) who additionally delineated the mechanism which involves Exo1 inducing the rapid release of ADP-ribosylation factor (ARF) preventing its GDP/GTP exchange. Normally, ARFs are required for membrane trafficking of vesicles to the plasma membrane; however, at high concentrations, Exo1 depletes its ability to exchange GDP and effectively halts its ability to traffic vesicles. Exo1 is currently sold by at least seven biotechnology retailers including Sigma Aldrich, Tocris, and Abcam. Some notable examples of its use in peer-reviewed articles include work by Eppler et al. 2017 (<https://www.ncbi.nlm.nih.gov/pmc/articles/PMC5342215/>) who used Exo1 to study the effect of exocytosis in the adhesion properties of T lymphocytes and Greene et al. 2017 (<http://jpet.aspetjournals.org/content/362/1/177.long>) who used Exo1 to suppress the constitutive exocytosis of the surface transport of a genetically mCherry-tagged Kv_{7.2} channel which 1) decreased the surface expression of mCherry on cells and 2) decreased the relative Kv_{7.2} current which was reversed following a media wash.

While we agree that many more experiments may be performed to further corroborate that MMP-2 is exocytosed into the extracellular space to degrade chondroitin sulfate proteoglycans, it is widely known that proteases in general and MMP-2 specifically are exocytosed by especially motile cells (<https://www.ncbi.nlm.nih.gov/pubmed/17981679>). The experiments, however, needed to thoroughly pinpoint a widely known mechanism would require extensive genetic

mutations in a cell system that would target the specific machinery involved in exocytosis, and we think, this is beyond the scope of this paper. We have, however, amended the text to clarify the use and limitations of the small molecule Exo1 inhibitor.

2.9 The data on MMP2 is exclusively based on pharmacological inhibitors, which could be unspecific. Genetic experiments for example using shRNA are needed for this conclusion.

The reviewer has expressed a concern with the use of MMP-2 inhibitors. Specifically, that “the data on MMP-2 is exclusively based on pharmacological inhibitors which could be unspecific.” Per the reviewer’s suggestion, we have performed additional experiments using shRNA-mediated knockdown of MMP-2. To begin, shRNA constructs targeting MMP-2 using lentiviral particle delivery to cells were validated in OPC cultures using western blotting controlled with an shRNA construct expressing GFP, which was also delivered using lentiviral particles. The shRNA construct was then applied in conjunction with or without ISP to assess the maturation and migration of OPCs on a substrate of chondroitin sulfate proteoglycans. While ISP+control lentiviral treatment alone was able to increase the area of MBP⁺ OPCs cultured on a low concentration of aggrecan, MMP-2 knock out attenuated the affects of ISP. Similar effects were seen in a migration assay where OPCs treated with ISP and shRNA knock down of MMP-2 showed a decrease in the ability to migrate past a highly concentrated aggrecan chondroitin sulfate proteoglycan barrier compared to ISP + control lenti-particle alone. Using a cerebellar slice model to assess remyelination after LPC-mediated demyelination, we applied the same constructs and found that ISP-mediated remyelination was indeed dependent on MMP-2 activity as shRNA knockdown of MMP-2 decreased remyelination of axons assessed through MBP and NF200 co-staining respectively.

To validate these constructs *in vivo* in order to assess the contribution of MMP-2 to ISP treatment, we returned to an LPC-induced model of demyelination of the dorsal columns of the spinal cord. MMP-2 immunostaining of the non-injured dorsal column showed slight diffuse staining of the white matter area. Following LPC intraspinal injections into the area, MMP-2 expression was slightly increased, however, MMP-2 expression was most strongly up-regulated in the demyelinated region following ISP treatment. Western blotting for MMP-2 of the lysate collected from the LPC-injected spinal cord confirmed an up-regulation of MMP-2 following ISP treatment as normalized by β -actin control. Additional immunohistochemistry of the ISP-treated dorsal column of 14 days post lesioned spinal cords revealed colocalization of MMP-2 with the oligodendrocyte-specific Olig2 marker. Finally, to assess whether remyelination is enhanced by MMP-2 activity following ISP treatment *in vivo*, we performed myelin staining using eriochrome cyananine of different treatment groups at 18 days post lesion. Interestingly, pharmacological inhibition of MMP-2 alone significantly increased the lesion volume compared to the LPC injected group. Compared to ISP treatment alone, which significantly decreased the lesion volume compared to the LPC control group, the addition of the MMP-2 inhibiting drug increased the lesion volume. Likewise, shRNA-mediated knock down of MMP-2 was able to decrease the efficacy of ISP in decreasing the lesion volume.

All together, these results validate that ISP, at least in part, enhances myelination regeneration through MMP-2 activity *in vivo* in a model of focal demyelination by LPC intraspinal injection

perhaps by enhancing OPC maturation and migration despite the presence of chondroitin sulfate proteoglycans up-regulated following injury.

Reviewers' comments:

Reviewer #1 (Remarks to the Author):

I authors have addressed some the issues pertaining to this article. However, several other issues remain unaddressed or unresolved:

Comment 1-2: I would like to point out that the cells in Figure 4D look like fibroblasts. Yes, the protocol you are using is supposed to yield 95% purity. That does not mean that it operates that way in your hands. Based on the dubious morphology of the "OPCs" in Figure 4D, more needs to be done to convince this reviewer. Also, the O4 staining is not convincing and needs additional controls.

Comment 1-5a: EAE is not considered a remyelination model and the examples that were given regarding injection of IL-10 expressing neural stem cells is not relevant. Also, this M1/M2 microglia explanation does not make any sense and I'm not sure why its is even included in the response. Yes, there could be other mechanisms at play between EAE and LPC (this is likely), but the manuscript does not dive into M1/M2 (which is also a dubious phenomenon) biology. I consider this matter completely unresolved.

Comment 1-5b: Just because other papers INFER that MMP2 is expressed in OPCs, does not mean it is a proven fact. The authors really need to show MM2 expression in OPCs in human MS lesions. This is critical and will only serve to enhance their manuscript.

Comment 1-5c: The shRNAi ex vivo slice experiment in Fig. 7 is very nice. However the in vivo LPC experiment is not convincing at all. The difference in lesion size could be variation between animals, etc. Need to show changes in proliferation, OPC markers, MBP, PLP, GFAP, etc.

Reviewer #2 (Remarks to the Author):

The authors have addressed most of the comments raised by both reviewers

Response to Reviewers:

Reviewer #1 raised 4 points

1.1 Comment 1-2: I would like to point out that the cells in Figure 4D look like fibroblasts. Yes, the protocol you are using is supposed to yield 95% purity. That does not mean that it operates that way in your hands. Based on the dubious morphology of the "OPCs" in Figure 4D, more needs to be done to convince this reviewer. Also, the O4 staining is not convincing and needs additional controls.

Answer: We thank the reviewer for raising this concern. To address this important issue of cell purity, we further identified the cells with additional well characterized oligodendrocyte cell lineage markers. Thus, we have performed additional immunostaining of NG2, Olig2 and PDGFR α in purified OPC cultures using the standard protocol in the lab. The results indicate that, indeed, the purity of OPCs exceeds 95% in our lab (Sup. Fig.7A). The reviewer points out that the cell morphology resembles that of fibroblasts. However, it is very important to address that cells typically alter their morphology in response to the inhibitory effects of CSPGs and we show and quantify this limited expansion of OPC cellular processes in several figures within the manuscript. However, to further address this concern, we have performed further spot assay experiments and immunostained the cells with additional OPC markers of NG2 and Olig2. More NG2⁺ and Olig2⁺ cells infiltrated and crossed the aggrecan-enriched outer-rim of the gradient spot after ISP peptide treatment. Also, the entire proteoglycan rim was reduced in intensity and diameter in the presence of ISP (Sup. Fig.7B).

1.2 Comment 1-5a: EAE is not considered a remyelination model and the examples that were given regarding injection of IL-10 expressing neural stem cells is not relevant. Also, this M1/M2 microglia explanation does not make any sense and I'm not sure why its is even included in the response. Yes, there could be other mechanisms at play between EAE and LPC (this is likely), but the manuscript does not dive into M1/M2 (which is also a dubious phenomenon) biology. I consider this matter completely unresolved.

Answer: EAE is commonly considered as an immune-mediated inflammatory demyelinating animal model. Many studies have demonstrated that active induced inflammation dynamically results in demyelination in the EAE model (Miller and Karpus, 2007; Nikic et al., 2011;

Ransohoff, 2012; Stanley and Pender, 1991). To date, although there is no perfect human MS animal model for studying remyelination, a number of identified myelin regenerative compounds, which do promote precocious differentiation of oligodendrocytes and attenuate clinical symptoms, have been tested for their ability to induce remyelination in the EAE model (Liu et al., 2016; Mei et al., 2016; Deshmukh et al., 2013; Najm et al., 2015; Samanta et al., 2015). Thus, while EAE is not a spontaneously remyelinating model, a number of labs have employed EAE as a model where therapeutic remyelination can occur. A recent study also indicated that accelerated remyelination supports axonal integrity and neuronal function after inflammatory demyelination in an EAE model, and that newly formed myelin remains stable at the height of inflammation due in part to the absence of MOG expression in immature myelin (Mei et al., 2016. eLife). Thus, we have chosen the EAE model as one test of the therapeutic effects of ISP on functional recovery and remyelination. Importantly, we show in Supp fig 4 e that early on in the presence of ISP, the initial demyelinating events due to EAE are equal in both control and treated animals. Thus, clearly ISP is not myelin degeneration protective. Since initial damage is equally severe, the recovery of myelin at later stages following ISP therapy must, therefore, be due to remyelination. Also, we show that the peptide is effective in the LPC model which does create a spontaneously, albeit slowly, remyelinating lesion.

In regards to the M1/M2 macrophage story, we highlight repeatedly in the manuscript in both the EAE and LPC models, the close spatial relationship between aggrecan and macrophages but also, importantly, the changes that occur in this association following treatment with ISP. We have also revealed changes in classic M1 versus M2 markers following peptide treatment. Thus, while the biological effects of modulation of the PTP σ receptor on microglia/macrophages still need to be elucidated, we have added further discussion of the role of ISP in modulating the inflammatory environment and the behavior of peptide treated microglia/macrophages on the evolution of CSPGs in the MS lesion. We also now cite and discuss recent work from our group (Dyck et al., 2018) that has reported that modulation of LAR family receptor phosphatases with synthetic peptides, including ISP, skewed microglia/macrophages towards an M2 polarization following spinal cord injury.

List of the references cited in 1.2:

- (1) Liu J, Moyon S, Hernandez M, Casaccia P. Epigenetic control of oligodendrocyte development: adding new players to old keepers. *Curr Opin Neurobiol.* 2016 Aug; 39():133-8.
- (2) Mei F, Mayoral SR, Nobuta H, Wang F, Despons C, Lorrain DS, Xiao L, Green AJ, Rowitch D, Whistler J, Chan JR. Identification of the Kappa-Opioid Receptor as a Therapeutic Target for Oligodendrocyte Remyelination. *J Neurosci.* 2016 Jul 27; 36(30):7925-35.
- (3) Deshmukh VA, Tardif V, Lyssiotis CA, Green CC, Kerman B, Kim HJ, Padmanabhan K, Swoboda JG, Ahmad I, Kondo T, Gage FH, Theofilopoulos AN, Lawson BR, Schultz PG, Lairson LL. A regenerative approach to the treatment of multiple sclerosis. *Nature.* 2013 Oct 17; 502(7471):327-332.
- (4) Najm FJ, Madhavan M, Zaremba A, Shick E, Karl RT, Factor DC, Miller TE, Nevin ZS, Kantor C, Sargent A, Quick KL, Schlatzer DM, Tang H, Papoian R, Brimacombe KR, Shen M, Boxer MB, Jadhav A, Robinson AP, Podojil JR, Miller SD, Miller RH, Tesar PJ. Drug-based modulation of endogenous stem cells promotes functional remyelination in vivo. *Nature.* 2015 Jun 11; 522(7555):216-20.
- (5) Samanta J, Grund EM, Silva HM, Lafaille JJ, Fishell G, Salzer JL. Inhibition of Gli1 mobilizes endogenous neural stem cells for remyelination. *Nature.* 2015 Oct 15; 526(7573):448-52.
- (6) Miller SD, Karpus WJ. Experimental autoimmune encephalomyelitis in the mouse. *Curr Protoc Immunol.* 2007 May; Chapter 15(): Unit 15.1.
- (7) Nikić I, Merkler D, Sorbara C, Brinkoetter M, Kreutzfeldt M, Bareyre FM, Brück W, Bishop D, Misgeld T, Kerschensteiner M. A reversible form of axon damage in experimental autoimmune encephalomyelitis and multiple sclerosis. *Nat Med.* 2011 Apr; 17(4):495-9.
- (8) Ransohoff RM. Animal models of multiple sclerosis: the good, the bad and the bottom line. *Nat Neurosci.* 2012 Jul 26; 15(8):1074-7.
- (9) Stanley GP, Pender MP. The pathophysiology of chronic relapsing experimental allergic encephalomyelitis in the Lewis rat. *Brain.* 1991 Aug; 114 (Pt 4):1827-53.
- (10) Mei F, Lehmann-Horn K, Shen YA, Rankin KA, Stebbins KJ, Lorrain DS, Pekarek K, A Sagan S, Xiao L, Teuscher C, von Bydingen HC, Wess J, Lawrence JJ, Green AJ, Fancy SP,

Zamvil SS, Chan JR. Accelerated remyelination during inflammatory demyelination prevents axonal loss and improves functional recovery. *Elife*. 2016 Sep 27;5.

1.3 Comment 1-5b: Just because other papers INFER that MMP2 is expressed in OPCs, does not mean it is a proven fact. The authors really need to show MMP2 expression in OPCs in human MS lesions. This is critical and will only serve to enhance their manuscript.

Answer: The reviewer raised an interesting issue but we suggest that a full analysis of MMP-2 in human MS subjects is far beyond the scope of the current manuscript which focuses on animal models of the disease. We hope in the future to investigate the effects of ISP on myelination/remyelination using human iPSCs and in human MS to show that the phenomena we detail in rodents is also occurring in humans. The potential inhibitory role of CSPGs in preventing remyelination in human MS has been documented.

1.4 Comment 1-5c: The shRNAi ex vivo slice experiment in Fig. 7 is very nice. However the in vivo LPC experiment is not convincing at all. The difference in lesion size could be variation between animals, etc. Need to show changes in proliferation, OPC markers, MBP, PLP, GFAP, etc.

Answer: As the reviewer suggested, we have performed and added the immunostaining of CS56, Olig2 and MBP of the spinal cord of MMP-2 shRNA-treated mice in Sup. Figure 11. We found that :1) shRNA knockdown of MMP-2 attenuated ISP-mediated digestive effects on CSPGs (Sup. Fig. 11A,D). 2) shRNA knockdown of MMP-2 decreased ISP-mediated accumulation of Olig2+ OPCs in the lesion epicenter (Sup. Fig. 11B,E), and 3) shRNA knockdown of MMP-2 attenuated the ability of ISP to enhance MBP expression in LPC-lesioned animals (Sup. Fig. 11C,F).

Reviewer #2 has no concerns or question raised.

REVIEWERS' COMMENTS:

Reviewer #1 (Remarks to the Author):

Authors have adequately addressed my concerns.